# Spatiotemporal dynamics of multidrug resistant bacteria on intensive care unit surfaces

Alaric W. D'Souza [1,8], Robert F. Potter[1,8], Meghan Wallace[2], Angela Shupe[2], Sanket Patel[1,2], Xiaoqing Sun[1,2], Danish Gul[3], Jennie H. Kwon[4], Saadia Andleeb[3]*, Carey-Ann D. Burnham[2,4,5,6]* & Gautam Dantas [1,2,5,7]*

Bacterial pathogens that infect patients also contaminate hospital surfaces. These contaminants impact hospital infection control and epidemiology, prompting quantitative examination of their transmission dynamics. Here we investigate spatiotemporal and phylogenetic relationships of multidrug resistant (MDR) bacteria on intensive care unit surfaces from two hospitals in the United States (US) and Pakistan collected over one year. MDR bacteria isolated from 3.3% and 86.7% of US and Pakistani surfaces, respectively, include common nosocomial pathogens, rare opportunistic pathogens, and novel taxa. Common nosocomial isolates are dominated by single lineages of different clones, are phenotypically MDR, and have high resistance gene burdens. Many resistance genes (e.g., $bla_{NDM}$, $bla_{OXA}$ carbapenamases), are shared by multiple species and flanked by mobilization elements. We identify _Acinetobacter baumannii_ and _Enterococcus faecium_ co-association on multiple surfaces, and demonstrate these species establish synergistic biofilms in vitro. Our results highlight substantial MDR pathogen burdens in hospital built-environments, provide evidence for spatiotemporal-dependent transmission, and demonstrate potential mechanisms for multi-species surface persistence.

[1] The Edison Family Center for Genome Sciences and Systems Biology, Washington University School of Medicine, St. Louis, MO, USA. [2] Department of Pathology and Immunology, Washington University School of Medicine, St. Louis, MO, USA. [3] Atta ur Rahman School of Applied Biosciences, National University of Sciences and Technology Islamabad, Islamabad, Pakistan. [4] Department of Medicine, Washington University School of Medicine, St. Louis, MO, USA. [5] Department of Molecular Microbiology, Washington University School of Medicine, St. Louis, MO, USA. [6] Departments of Pediatrics, Washington University School of Medicine, St. Louis, MO, USA. [7] Department of Biomedical Engineering, Washington University in St. Louis, St. Louis, MO, USA. [8] These authors contributed equally: Alaric W. D'Souza, Robert F. Potter. *email: saadiamarwat@yahoo.com; cburnham@wustl.edu; dantas@wustl.edu

Global treatment of bacterial infections is increasingly compromised by evolution and transmission of multidrug-resistant organisms (MDROs) and their anti-biotic resistance genes (ARGs) between multiple habitats[1]. Infections caused by MDROs are associated with increased mortality risk compared to infections by matched species susceptible isolates[2–4]. Through international travel, clonal expansion, and promiscuous mobile genetic elements, MDROs and the ARGs they harbor have rapidly swept across the globe[1,5–12]. Resistant infections cause over 23,000 annual deaths in the United States of America (USA) and cost the economy over 55 billion dollars[13]. The annual global death toll from MDROs is at least 700,000 people[14]. Improved surveillance and understanding of MDRO and ARG transmission are key factors in reducing these death tolls[1,12].

Hospitalized patients are more vulnerable to bacterial infections than the general population[15], and healthcare-associated infections (HAIs) acutely threaten patient safety worldwide[16,17]. The "ESKAPE" pathogens, named by the Infectious Disease Society of America, are common causes of HAIs and the most common MDROs[18]. These include the gram-positive micro-organisms *Enterococcus* spp. and *Staphylococcus aureus*, and the gram-negative microorganisms *Klebsiella pneumoniae*, *Acineto-bacter baumannii*, *Pseudomonas aeruginosa*, and *Enterobacter* spp[18]. These ESKAPE pathogens can be acquired while hospitalized, but some patients may be colonized or infected prior to hospital admission[19]. Patients harboring these putative pathogens can transmit these bacteria to healthcare workers, other patients, medical equipment, and hospital surfaces[19], but the relative contribution of this contamination route compared to other routes in unknown. The presence of these microorganisms on surfaces in healthcare settings is a local and global public health concern[20]. Some putatively pathogenic strains of bacteria persist for months on hospital surfaces, and they may even survive surface decontamination efforts, partly aided by biofilm formation[21–24]. Though studies clearly demonstrate that bacterial pathogens exist on hospital surfaces, key knowledge gaps exist regarding the levels, types, and dynamics of contamination in hospitals from different geographies[15,19]. Specifically, there is a lack of information on the spatial, temporal, and phylogenetic relationships between different bacterial taxa on surfaces from countries endemic for a high burden of ARGs. This information gap is especially true for physical colocalization and horizontal gene transfer between clinically relevant ESKAPE pathogens and benign environmental bacteria.

Monitoring high contact surfaces for clinically relevant pathogenic bacteria and understanding the dynamics of their persistence and spread is one approach to thwart MDRO transmission and protect vulnerable hospitalized patients[25]. Additionally, such surveillance provides an opportunity to identify and characterize potential emerging pathogens before they are recognized in clinical infections[13,26].

To address the question of MDRO spatiotemporal dynamics and persistence on healthcare surfaces we conducted a year-long longitudinal study at a tertiary care hospital in Pakistan (PAK-H) where endemic ARG burden is high[27–29]. Differing resistance mechanisms to last-resort carbapenem antibiotics have been found in genetically similar *Enterobacteriaceae* strains and plasmids isolated from hospitals in Pakistan and the USA[30]. Accordingly, we included a matched tertiary care hospital in the USA (USA-H) as a comparison group. For our collections and subsequent analysis, we took an Eulerian approach by selecting and measuring fixed hospital surfaces over time to understand bacterial contamination dynamics. This approach allows us to leverage collection time information and surface spatial information to draw epidemiological insights. In both hospitals, we sampled four intensive care unit (ICU) rooms with five surfaces in each room (Fig. 1). We collected surface swabs every other week for 3 months, and again at 6 months, and at 1 year, for a total of 180 samples per hospital. We identified high burdens of known MDROs on PAK-H ICU surfaces including ESKAPE pathogens and novel taxa[31,32]. This investigation is the first to show such widespread contamination with multidrug-resistant, extensively drug-resistant, and pan-drug-resistant bacteria in Pakistan. We found evidence that bacteria are non-randomly distributed on hospital surfaces with respect to both space and time, and we used this information to narrow possible contamination routes. We found cross-contamination of MDRO clones both across different surfaces within rooms, as well as between rooms at the same sampling time-points. From our results, it is likely that bacteria are seeded to hospital surfaces from diverse human and/or environmental reservoirs in a time-dependent manner. These seedings result in waves of contamination that are often, but not always, restricted to a single collection time. We show high numbers of ARGs are shared between common nosocomial pathogens and rarer bacterial species, including several novel taxa, which are close phylogenetic relatives to nosocomial pathogens. Co-association analysis of *A. baumannii* and *Enterococcus faecium* led us to identify synergistic biofilm formation between these two ESKAPE pathogens. This discovery points to a possible explanation for multi-species bacterial persistence on hospital surfaces. Longitudinal persistence of these high-impact pathogenic species alongside highly resistant bacteria classically identified as "environmental" paints a concerning picture of hospital surface contamination. These results lay the groundwork for future surveillance efforts and infection

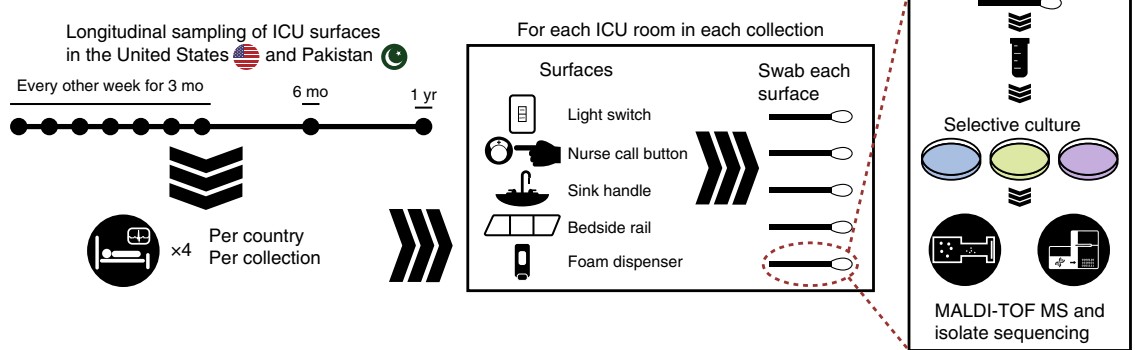

**Fig. 1** Overview of sample collection and processing. Samples were collected from surfaces longitudinally over the course of 1 year from PAK-H ICU and USA-H ICU. Four rooms from each ICU were chosen for sampling and five surfaces within each room were surveyed for every collection time. Bacteria were cultured from the collection swabs, identified by MALDI-TOF MS, and then whole-genome sequenced

control interventions to reduce healthcare associated bacterial surface contamination.

## Results

**PAK-H ICU surfaces had high bacterial burden**. We recovered 1163 bacterial isolates from hospital surfaces in PAK-H and predicted their species identities by matrix-assisted laser desorption/ionization time-of-flight mass spectrometry (MALDI-TOF MS). We chose a subset of 289 unique isolates for phenotypic and genomic analysis, using the criterion of a single isolate per unique MALDI-TOF MS identified species, per culture condition, per surface, per time-point (Supplementary Data 1). These 289 bacteria represent 31 species and 10 families (Fig. 2a). A total of 25.9% (75/289) of isolates recovered from PAK-H were identified as *A. baumannii*. A total of 16.2% (47/289) were the gram-positive pathogen *E. faecium*, and 11.8% (34/289) were *K. pneumoniae*. Interestingly, similar numbers of the soil-associated opportunistic pathogen *Pseudomonas stutzeri* were recovered (28/289, 9.7%) as the common nosocomial pathogen *P. aeruginosa* (27/289, 9.3%). In addition to these expected nosocomial organisms, we identified a variety of other clinically relevant species such as *Stenotrophomonas maltophila*, *Shewanella putrefaciens*, and *Providencia rettgeri*. These results starkly contrast with USA-H, where we only recovered six unique isolates, which MALDI-TOF MS identified as *A. baumannii* (4/6) and *E. coli* (2/6) (Fig. 2a). The majority of PAK-H (156/180, 86.7%) surface collections yielded bacteria (Fig. 2b), but only a few (6/180, 3.3%) USA-H surface collections yielded isolates using the same culture conditions.

**Sequence-based bacterial identification outperformed MALDI-TOF MS**. We performed draft Illumina whole-genome sequencing (WGS) on the 289 isolates to improve taxonomic resolution, quantify transmission dynamics for abundantly recovered organisms, and analyze ARG content (Supplementary Table 1). Initially, we constructed a Hadamard matrix, which represents the product of the average nucleotide identity (ANI) and percent of the genome aligned, between every pairwise combination of the 289 genomes sequenced from PAK-H surfaces (Supplementary Fig. 1). Hierarchical clustering of Hadamard values confirms 74/75 isolates identified by MALDI-TOF MS as *A. baumannii*, 47/47 as *E. faecium*, 33/34 as *K. pneumoniae*, 27/27 as *P. aeruginosa*, and 24/28 as *P. stutzeri*. These isolates cluster into the first five blocks of the matrix. Analysis of the clustering pattern in the *K. pneumoniae* group found one isolate distant from the rest of the cohort; separate ANI analysis demonstrated this isolate is *Klebsiella quasipneumoniae*. Similarly, three isolates annotated as *P. stutzeri* are *Pseudomonas xanthomarina*. The isolate identified as *A. baumannii* that did not cluster with the rest of the cohort was *Acinetobacter soli*. In total, we found 27 cases where initial MALDI-TOF MS identifications differed from subsequent WGS-dependent identifications. Additionally, both (2/2) isolates initially identified as *Empedobacter brevis* are *Empedobacter falsenii*. Two out of three of genomically confirmed *Atlantibacter subterranea* were unidentified by MALDI-TOF MS but 1/3 was identified as the closely related *Atlantibacter hermanii*.

We found 12 instances where genomes did not have >95% ANI with the identified MALDI-TOF MS hit or the most closely related genomes as determined by 16S rRNA gene sequence in the EzBioCloud database, indicating that these are putative novel genomospecies. A separate investigation found that 2/7 of the isolates unidentified by MALDI-TOF MS are a new genus of multidrug-resistant *Enterobacteriaceae*, termed *Superficieibacter electus*[31]. The previously unreported genomospecies come from the *Caulobacteriacae*, *Xanthomonadaceae*, and *Enterobacteriaceae*

families, and five of the proposed new genomospecies are *Pseudomonadaceae*. Importantly, these unreported genomospecies are found on the same healthcare surfaces as common human pathogens. Our results indicate WGS offers improved resolution for species delineation compared to conventional clinical diagnostic tools, for both common human pathogens and rarer species.

**Single lineages dominated *A. baumannii* and *E. faecium* populations**. As our taxonomic analysis demonstrated *A. baumannii*, *E. faecium*, *K. pneumoniae*, and *P. aeruginosa* were the most abundant putative pathogens collected at PAK-H, we next endeavored to determine population structure for isolates in these species. For each species, we annotated protein coding sequences with Prokka, constructed core genome maximum-likelihood phylogenetic trees with Roary and RAxML, then identified lineages with fastGEAR/BAPS[33–36]. Our results demonstrate that for *A. baumannii* and *E. faecium*, but not *K. pneumoniae* or *P. aeruginosa*, a single lineage represented >70% of all isolates collected over 12 months. For all four species, time of collection, but not room or surface had the greatest concordance with phylogenetic position (Fig. 3).

A total of 88.4% (69/78) of the *A. baumannii* isolates were from lineage 7 (Fig. 3a, Supplementary Fig. 1a), which was composed of several untypable isolates, and 7 sequence types (STs). Interestingly, the four USA-H genomes in ST208 clustered adjacent to one another and next to the seven ST208 genomes from PAK-H. A total of 72.3% (34/47) of the *E. faecium* isolates come from BAPS lineage 4. All lineage 2 and lineage 1 *E. faecium* isolates came from the second and fourth week, respectively. *K. pneumoniae* contained five BAPS lineages with ST617, ST337, ST231, and ST147 relating to lineages 1, 2, 4, and 5, respectively. All the lineage 2 *K. pneumoniae* came from week 4 of our collections. *P. aeruginosa* had the greatest concordance between lineages and sequence types, as ST859, ST664, ST235, and ST571 corresponded to lineages 1, 2, 3, and 4, respectively. Seventy-four percent (20/27) of the *P. aeruginosa* isolates came from week 8 of our collection, including all lineage 4 and lineage 1 isolates. Our analysis of population structure for recovered *A. baumannii*, *E. faecium*, *K. pneumoniae*, and *P. aeruginosa* indicates that specific lineages of closely related isolates dominated PAK-H surfaces. We next wanted to investigate if clonal groups of highly related isolates existed within lineages we identified for these pathogens.

**Genetically related isolates are spatially and temporally linked**. To identify clonality within bacterial species, we removed recombinant positions from the core genome alignment (Supplementary Data 2) and calculated pairwise single nucleotide polymorphism (SNP) distances (Supplementary Fig. 2, Supplementary Data 3). All four bacterial species had multimodal pairwise SNP distance distributions, indicating concordance between SNP distance and clustering on core genome phylogenetic trees (Supplementary Fig. 3). To investigate potential transmission events in PAK-H surface collections, we used our pairwise SNP distances to find clonal bacterial isolates. Clonal groups were defined conservatively as isolates of the same species with five or fewer core genome SNPs between any two members of the group. We identified potential clonal groups within *A. baumannii* and *E. faecium* (Supplementary Data 4). *A. baumannii* had 11 clonal groups encompassing 45 of the 79 (57.0%) *A. baumannii* isolates (Fig. 4a). All these *A. baumannii* clonal groups were in BAPS lineage 7. *E. faecium* had six clonal groups from four different lineages (Fig. 4b). Foty-one out of 48 (85.4%) *E. faecium* isolates belonged to one of the six clonal groups. Principal coordinates analysis and PERMANOVA for *A.*

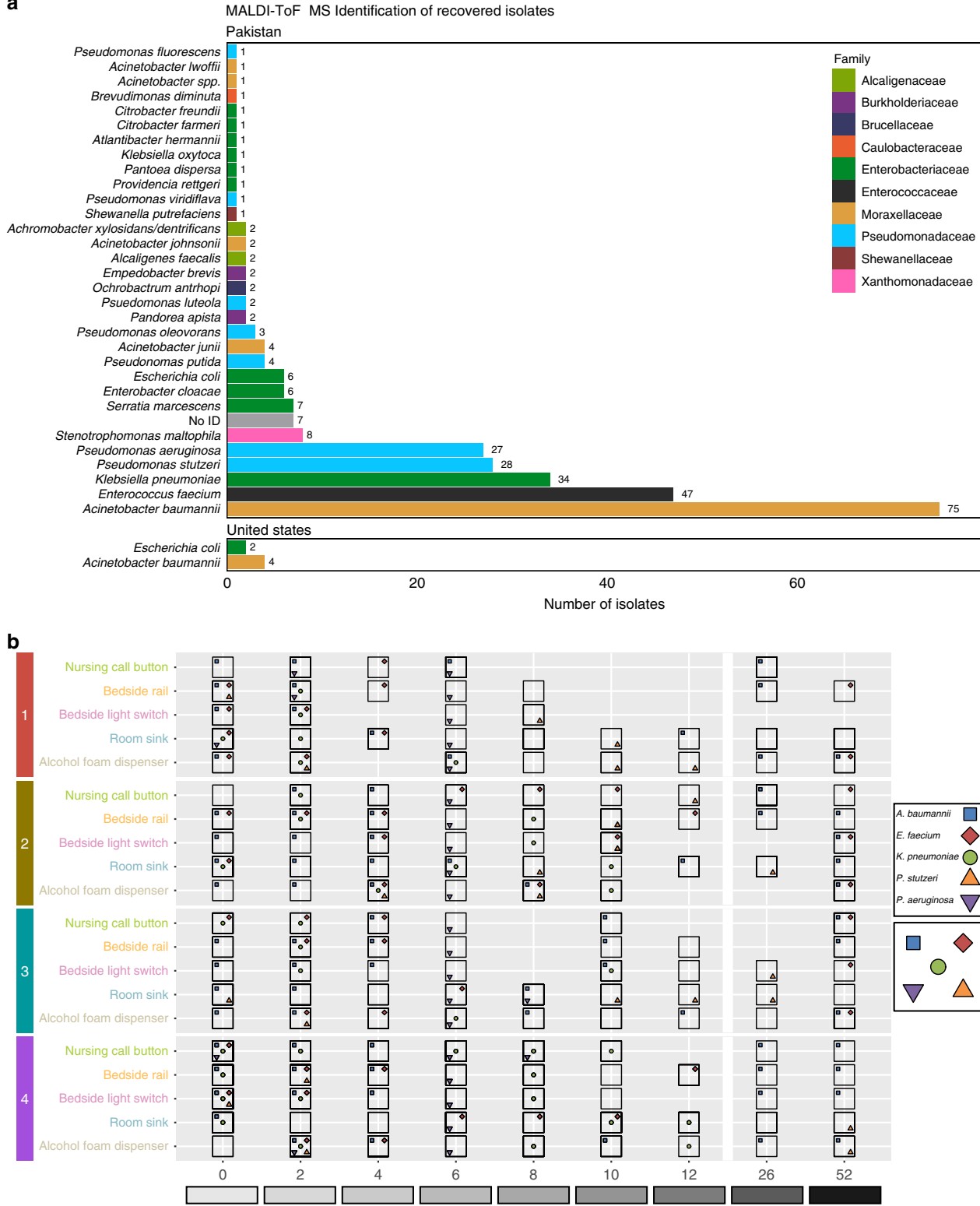

**Fig. 2** Bacterial isolate taxonomic identification and location. **a** MALDI-TOF MS identifications of bacterial isolates recovered from surfaces at PAK-H (above) and USA-H (below), colored by family. **b** Overview of PAK-H bacterial surface collections. Each horizontal gray panel represents a PAK-H room. Within each room, the horizontal gridded white lines are the five sampled surfaces. Each vertical white line is one of the collection weeks. Places where the horizontal and vertical white lines intersect represent a sampling. Large, open black boxes are around any surface where one or more bacteria were collected. The five most abundant species in the collections are indicated as colored shapes within the black boxes of surfaces where they were collected (blue squares are *A. baumannii*, red diamonds are *E. faecium*, green circles are *K. pneumoniae*, orange triangles are *P. stutzeri*, and purple triangles are *P. aeruginosa*). Source data for both panels are provided in the source data file

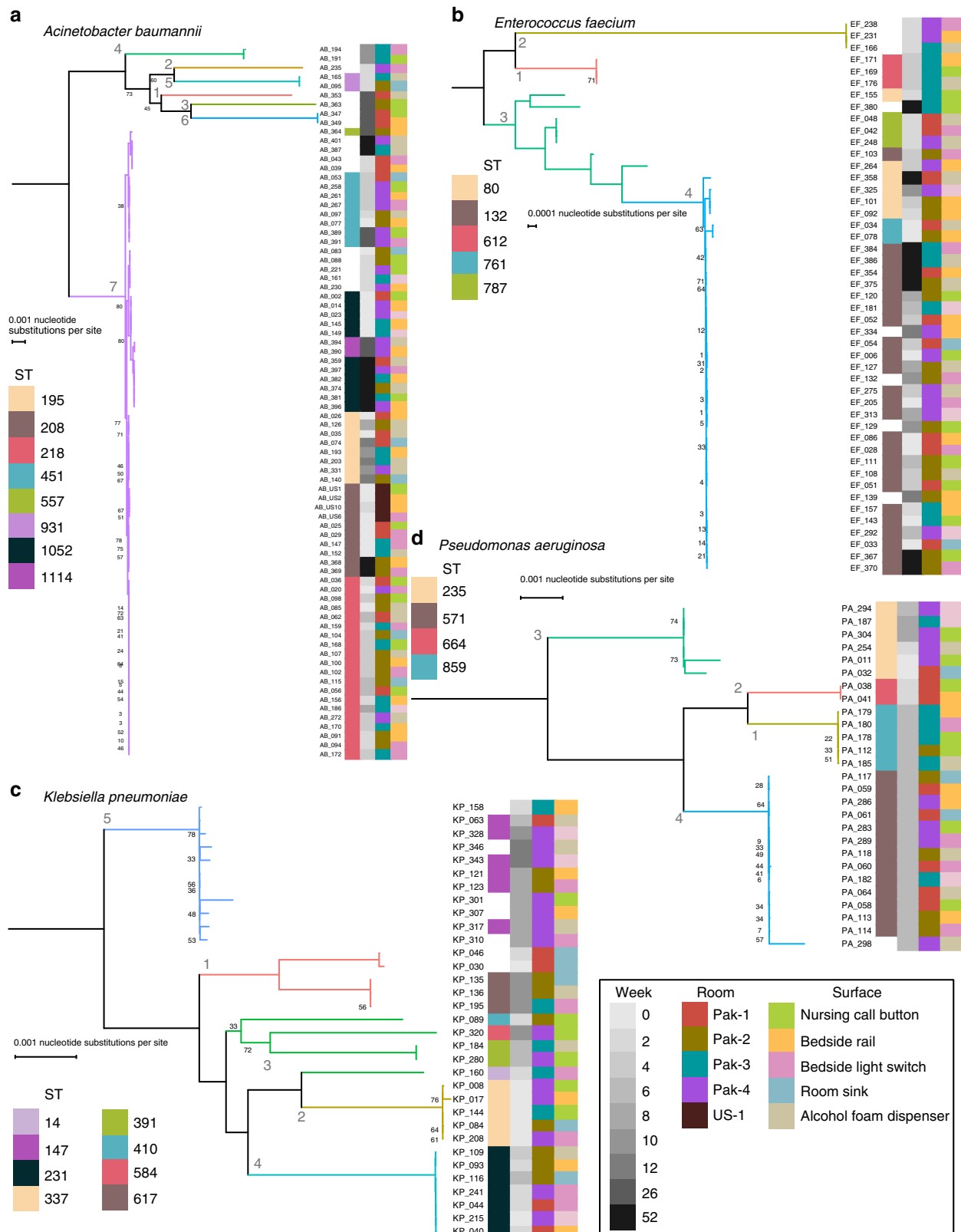

**Fig. 3** Phylogenetic trees of high abundance species from core genome alignments. Maximum likelihood phylogenetic trees from core genome alignments of *A. baumannii* (**a**), *E. faecium* (**b**), *K. pneumoniae* (**c**), and *P. aeruginosa* (**d**). Tree branches are colored by hierBAPS lineage and these lineages are colored in subsequent figures. Sequence type, week, room, and surface are annotated as colored bars next to the isolate number. Week is given as grayscale with darker values corresponding to later weeks. The US room that yielded isolates is annotated dark brown. The Pakistan rooms are red, olive, turquoise, and purple for rooms 1–4, respectively. Surfaces are lime green for nursing call button, light orange for bedside rail, pink for bedside light switch, blue for room sink, and light brown for alcohol foam dispenser. Scale bars for nucleotide substitutions per site are indicated for each tree. Source data for all panels are provided in the source data file

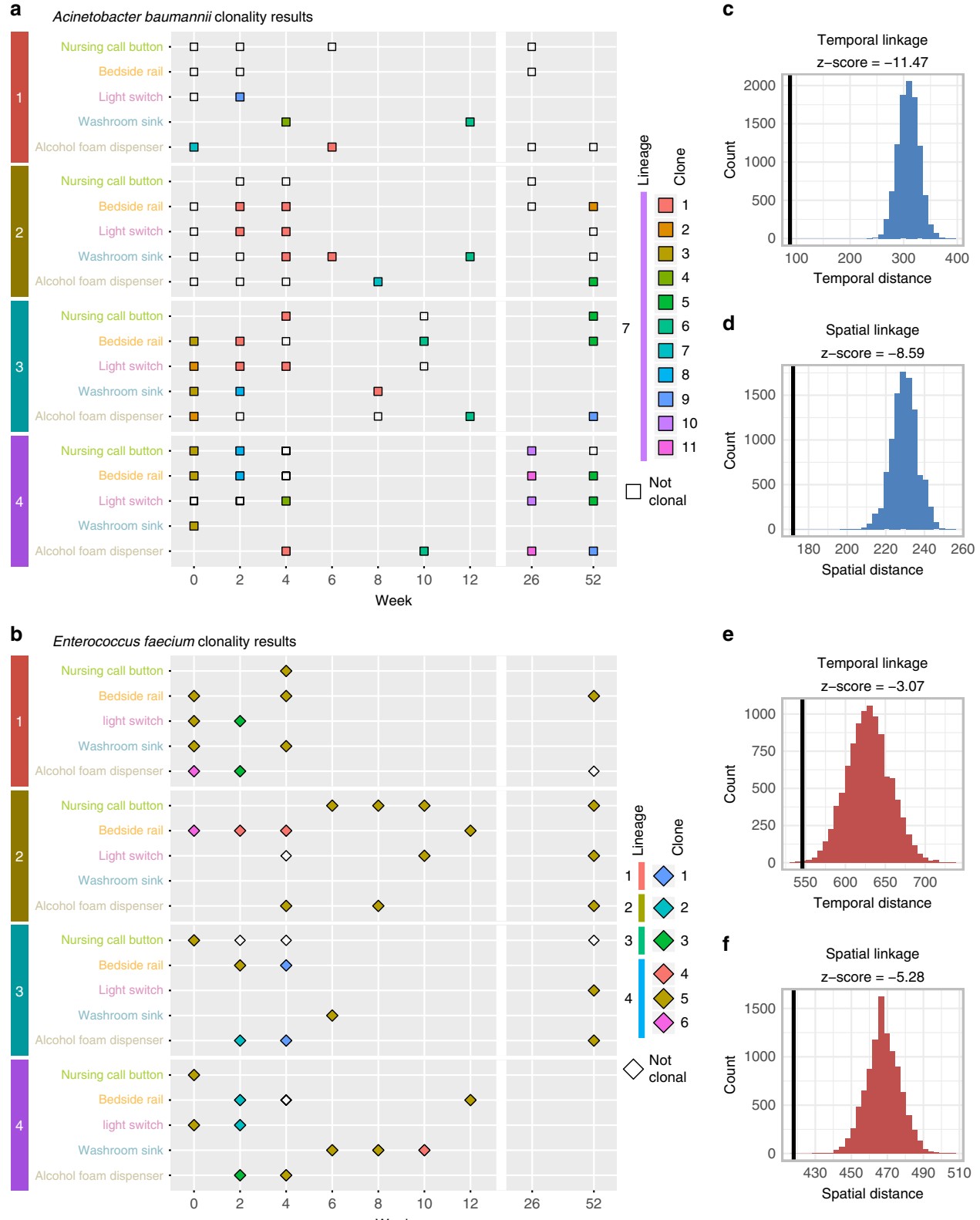

baumannii and *E. faecium* clonal groups and non-clonal isolates indicated bacteria from the same clonal group also had significantly similar accessory gene content with ($P = 0.0093$ for *A. baumannii* and $P < 0.0001$ for *E. faecium* (PERMANOVA)) and without ($P < 0.0001$ for both (PERMANOVA)) non-clonal isolates included (Supplementary Fig. 4). Together, these results indicate clonal populations of *A. baumannii* and *E. faecium* exist

on PAK-H surfaces and that these clones could have a genetic or phenotypic basis for extended persistence.

We conducted a 10,000 trial, permutation test of the first 3 months of collection to determine if clonal isolates of *A. baumannii* and *E. faecium* were significantly spatially and temporally coincident by comparing the observed spatial or temporal distance with the imputed null distribution

**Fig. 4** Relationship of core genome SNP groups to spatial and temporal distance. **a** Clonality results for *A. baumannii*. Squares represent *A. baumannii* collected from surfaces. Colors represent clonal subgroup membership. Each colored set is a clonal subgroup with fewer than five SNPs different between all members of the group. Unfilled squares did not have fewer than five SNPs different with any other isolates. Lineage from BAP (identified in Fig. 3 by branch color) is indicated in the legend on the left. **b** Clonality results for *E. faecium*. Diamonds represent *E. faecium* collected from surfaces. Colors represent clonal subgroup membership. Each colored set is a clonal subgroup with fewer than five SNPs different between all members of the group. Unfilled diamonds did not have fewer than five SNPs different with any other isolates. Lineage from BAP is indicated in the legend on the left. For **c**, **d**, temporal distances are calculated as +1 for every 2-week span separating isolate collections. Spatial distances are given as +0 if isolates were collected from the same surface and room, +1 if they were collected from the same room, but different surfaces, and +2 if they were collected from different rooms. **c** Temporal linkage for *A. baumannii* clones. The expected temporal distance distribution is shown in blue and the observed temporal distribution is shown as a solid black line. **d** Spatial linkage for *A. baumannii* clones. The expected spatial distance distribution is shown in blue and the observed spatial distribution is shown as a solid black line. **e** Temporal linkage for *E. faecium* clones. The expected temporal distance distribution is shown in red and the observed temporal distribution is shown as a solid black line. **f** Spatial linkage for *E. faecium* clones. The expected spatial distance distribution is shown in red and the observed spatial distribution is shown as a solid black line. Source data for all panels are provided in the source data file

(Supplementary Fig. 5, Supplementary Data 5). Individual clone data were aggregated by species to determine if clones were grouped spatially and temporally with all other clones (Fig. 4c–f). These test results indicate clonal bacteria from both *A. baumannii* and *E. faecium* are more likely than predicted by chance to be isolated from surfaces that are spatially ($P < 0.00001$ for both *A. baumannii* and *E. faecium* (permutation test)) or temporally ($P < 0.00001$ for *A. baumannii* and $P = 0.0021$ for *E. faecium* (permutation test)) close together than they are to be isolated from surfaces that are spatially or temporally distant. This could represent a threat for clonal HAI outbreaks in patients linked by room or duration in the PAK-H ICU. Additionally, it is likely that reducing bacterial burden on specific surfaces would also reduce bacterial burden on linked surfaces.

**Variant distances correlate with core genome SNP distances**. Since core genome SNPs may underestimate true differences between isolates from exclusion of intergenic regions, we augmented our analysis by mapping quality filtered reads from *A. baumannii* and *E. faecium* to their respective type strain complete genome assemblies (GCF_000746645.1 (https://www.ncbi.nlm.nih.gov/assembly/GCF_000746645.1/) for *A. baumannii* and GCF_000174395.2 (https://www.ncbi.nlm.nih.gov/assembly/GCF_000174395.2/) for *E. faecium*). With these mapped reads, we called multiple different variant types including SNPs, multiple nucleotide polymorphisms (MNPs), and insertions/deletions (indels). These variant calls gave us high-resolution pairwise genetic distances between isolates of the same species (Supplementary Fig. 6).

We compared these variant distances to the core genome SNP distances (Supplementary Fig. 7). Correlation between pairwise core genome SNP distance and pairwise variant distance was significant for both *A. baumannii* ($R^2 = 0.9471$, $P < 0.0001$ (linear model)) and *E. faecium* ($R^2 = 0.8696$, $P < 0.0001$ (linear model)). These results suggest core genome SNP distances and variant distances are tightly linked in our cohort. Despite this tight correlation, higher resolution of variant distances compared to core genome SNP distances was apparent (Supplementary Fig. 7). Both variant and core genome SNP distances for *A. baumannii* are bimodal reflecting between lineage distances, but variant distances have a more continuous spread than core genome SNP distances (Supplementary Fig. 7a). While *E. faecium* was trimodal rather than bimodal for SNP distances, variant distances were again more continuous than core genome SNP distances (Supplementary Fig. 7b). We exploited this higher resolution to investigate biologically meaningful isolate groupings.

**Spatiotemporal distance identifies relevant epidemiologic groups**. To identify epidemiologically meaningful groupings, we leveraged space and time information from our collections. For *A. baumannii* and for *E. faecium*, we iterated through every unique variant distance cutoff from the lowest distance between any two isolates until the lowest distance between any two isolates not in the same lineage (Figs. 5a–e, 6a–e). We used these cutoffs to filter the isolate pairwise links edge list. For each cutoff, we found perfectly reciprocal groups with maximal graph coverage and recorded the number of cliques and the number of isolates per clique (Supplementary Fig. 8, Figs. 5a, b, 6a, b, Supplementary Data 6). Here we define cliques as complete subgraphs within the network where each node in the clique is connected to each other node in the clique. Both *A. baumannii* and *E. faecium* showed a similar pattern where number of cliques rises sharply initially and then peaks. During this peak, there is a gradual increase in the number of isolates per clique, with cliques staying relatively balanced. After peaking, the number of cliques rapidly declines as formerly independent cliques merge. This merging interestingly results in one major clique with several other minor cliques. We then determined how much each clique grouping's spatial and temporal distances deviated from a null model generated with 10,000 permutations for that clique grouping (Figs. 5c, d, 6c, d, Supplementary Data 6). If isolates spread randomly on surfaces, we would expect z-scores close to 0 for the spatial and temporal data. We projected the lowest z-score cutoffs onto the pairwise variant distances histogram (Figs. 5e, 6e). The greatest deviation from the null model for significant temporal (Figs. 5f, 6f) and spatial linkage (Figs. 5g, 6g) coincided with cutoffs that yielded the highest number of cliques. In this case, we found nine cliques for *A. baumannii* with both the time-minimizing distance (Fig. 5h) and space-minimizing distance (Fig. 5i) cutoff. For *E. faecium*, we found ten cliques for the time-minimizing cutoff (Fig. 6h) and eight cliques for the space-minimizing cutoff (Fig. 6i). The cutoff values in that range best fit the radiation of isolates on these surfaces. After cutoff values increase beyond the clique-maximizing value, within-clique spatial and temporal distance observations rapidly increase to match and even exceed null estimations, indicating that the epidemiologically relevant variant cutoff was likely passed.

For *A. baumannii* and for *E. faecium*, cliques are mostly restricted to single collection times, but some cliques, like clique 8 for *A. baumannii*, deviate from this trend and are instead broadly spread over surfaces in both time and space. Though most cliques are restricted by time, cliques that are spread in time show room restricted patterning. This distribution of isolates could be explained by a reservoir of multiple clones with continual seeding to surfaces. In this scenario, most seeding events would not result in long-term surface, persistence, but a few clones could pass this strong filter to successfully survive for multiple weeks within rooms in a space-dependent fashion.

**PAK-H isolates have high genotypic and phenotypic resistance**. We used ResFinder to identify ARGs in draft genomes of our

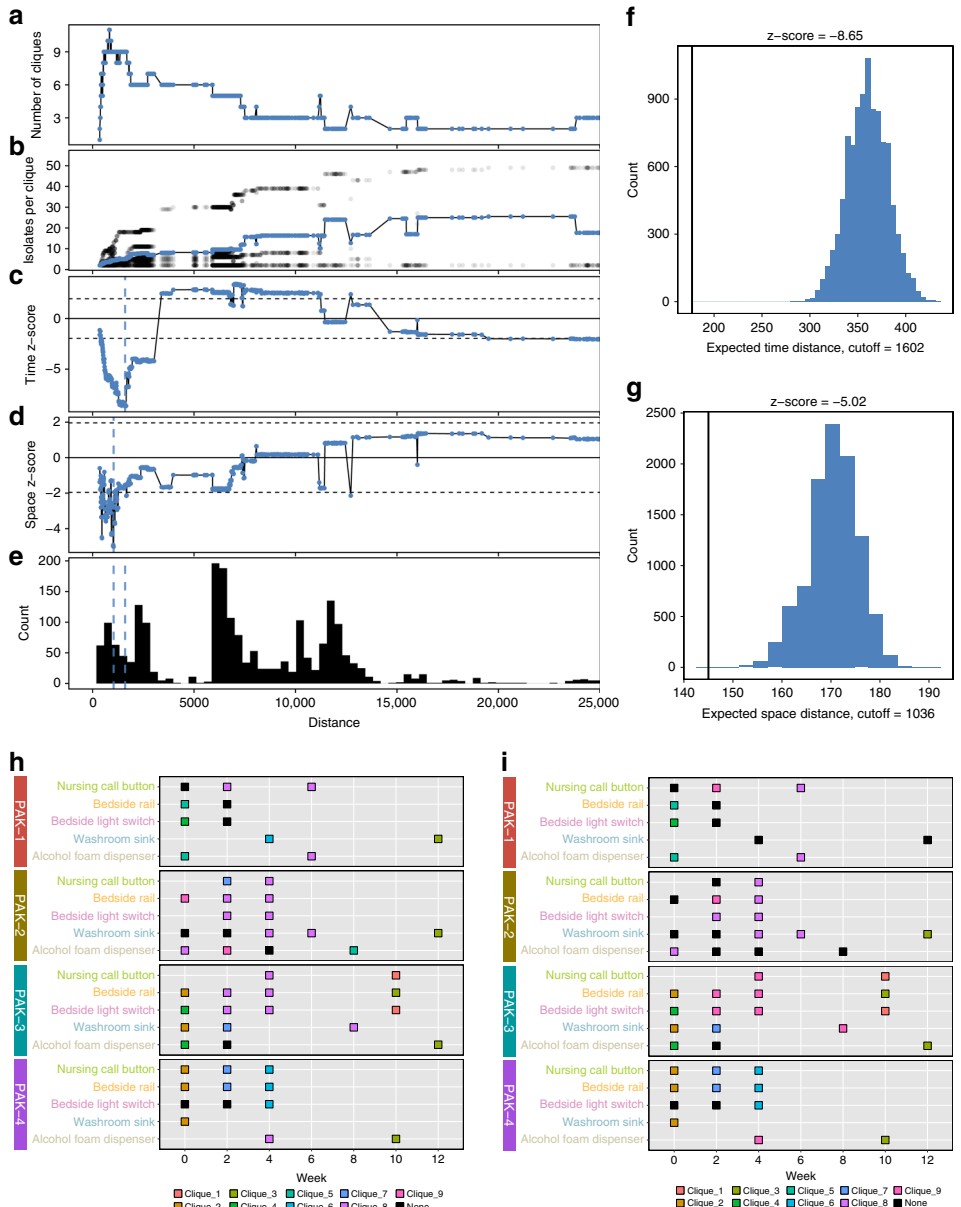

**Fig. 5** *A. baumannii* optimal pairwise variant distance cliques identified by spatial and temporal data. *A. baumannii* isolate clique groupings with the lowest observed spatial and temporal distances compared to the expected distribution. Panels **a–e** give information on the clique groupings for each unique variant distance cutoff (*x*-axis) starting at the minimum variant distance and extending until the minimum variant distance between two isolates from different lineages. Panel **a** shows the number of completely connected cliques identified at the current cutoff value. Panel **b** shows the number of isolates per clique, with the black dots showing each individual clique and the blue points showing the average per clique. Panels **c** and **d** show the deviation (*z*-score given as blue points) of the observed temporal (**c**) or spatial (**d**) distance compared to the expected distribution. The dotted lines show the upper and lower significance bounds. **e** Histogram of the number of pairwise comparisons in different variant distance cutoff ranges. Blue dashed lines show the minimum *z*-score cutoffs for the temporal and spatial distances given in **c** and **d**. Panels **f** and **g** show the observed distance value vs the expected distribution for the minimum *z*-score values identified in **c** and **d** for temporal (**f**) and spatial (**g**) distance. The black vertical line is the observed distance and blue filled histogram is the expected distribution. Panels **h** and **i** show the cliques identified at the minimum *z*-score cutoffs for temporal (**h**) and spatial (**i**) distance measurements. Source data for all panels are provided in the source data file

sequenced *A. baumannii*, *E. faecium*, *K. pneumoniae*, and *P. aeruginosa* isolates[37]. Additionally, we determined if these isolates were phenotypically resistant, intermediate, or susceptible using Kirby–Bauer disk diffusion assays (Supplementary Data 7) in accordance with Clinical and Laboratory Standards Institute (CLSI) guidelines[38]. For all species, we found hierarchical clustering of isolates based on ARG presence or phenotypic susceptibility indicated lineage was the major predictor of resistance-based clustering patterns (Fig. 7). Specific lineages can dominate

clinical infections and tight correlation of lineage with resistance may relate to this phenomenon[39]. This linkage between lineage and antimicrobial resistance may also allow for rapid sequence-based, rather than gene-based, susceptibility predictions[40].

*A. baumannii* isolates harbored 30 unique ARGs against nine different classes of antimicrobials (Fig. 7a). Forty percent (12/30) of these ARGs were β-lactamases and 26.7% (8/30) were expected to confer phenotypic resistance against aminoglycosides (Fig. 7a). One-hundred percent (65/65) of lineage 7 PAK-H isolates

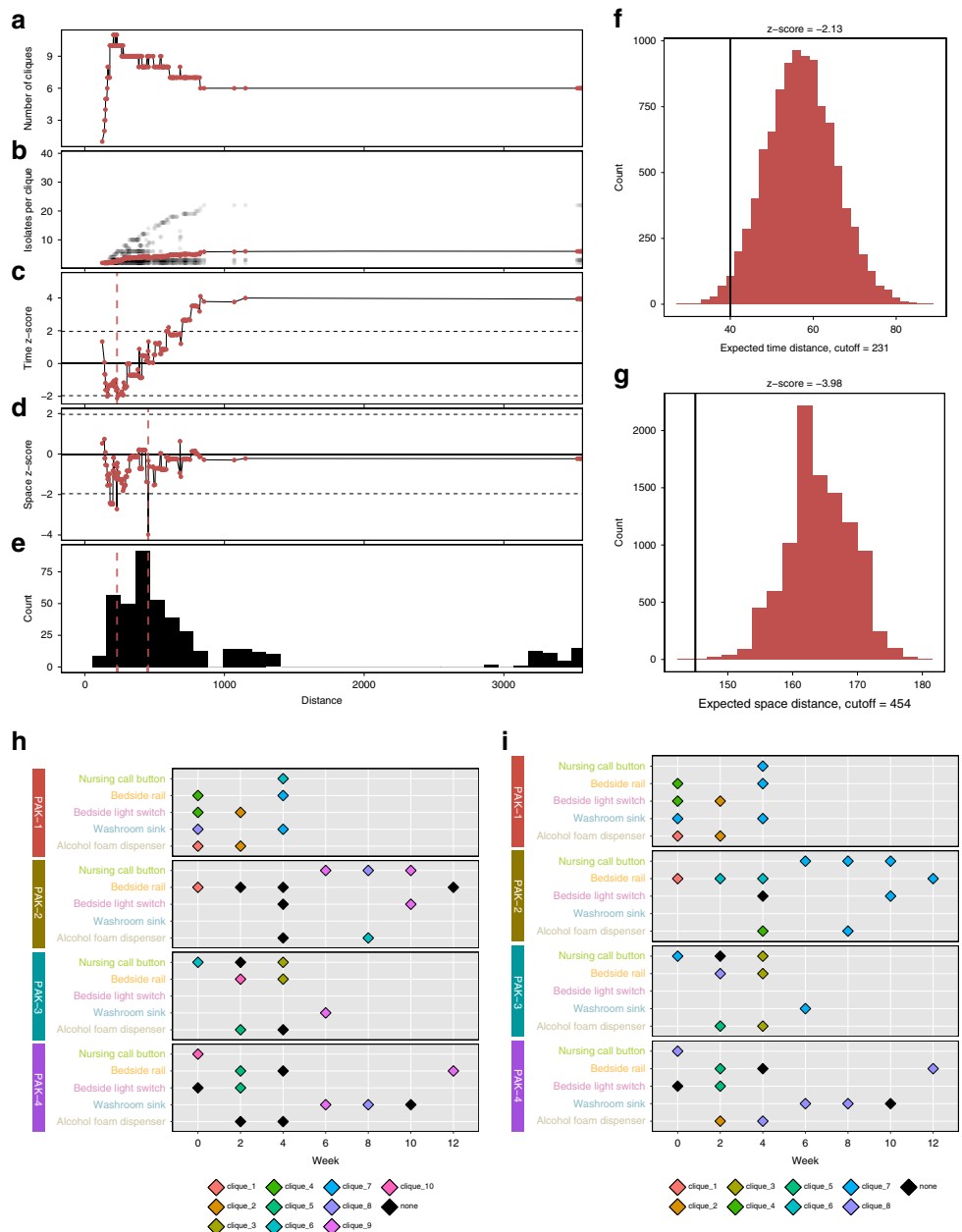

**Fig. 6** *E. faecium* optimal pairwise variant distance cliques identified by spatial and temporal data. *E. faecium* isolate clique groupings with the lowest observed spatial and temporal distances compared to the expected distribution. Panels **a–e** give information on the clique groupings for each unique variant distance cutoff (*x*-axis) starting at the minimum variant distance and extending until the minimum variant distance between two isolates from different lineages. Panel **a** shows the number of completely connected cliques identified at the current cutoff value. Panel **b** shows the number of isolates per clique, with the black dots showing each individual clique and the red points showing the average per clique. Panels **c** and **d** show the deviation (*z*-score given as red points) of the observed temporal (**c**) or spatial (**d**) distance compared to the expected distribution. The dotted lines show the upper and lower significance bounds. **e** Histogram of the number of pairwise comparisons in different variant distance cutoff ranges. Red dashed lines show the minimum *z*-score cutoffs for the temporal and spatial distances given in **c** and **d**. Panels **f** and **g** show the observed distance value vs the expected distribution for the minimum *z*-score values identified in **c** and **d** for temporal (**f**) and spatial (**g**) distance. The black vertical line is the observed distance and red filled histogram is the expected distribution. Panels **h** and **i** show the show the cliques identified at the minimum *z*-score cutoffs for temporal (**h**) and spatial (**i**) distance measurements. Source data for all panels are provided in the source data file

harbored $bla_{OXA-23}$ and 95.4% (62/65) also had $bla_{OXA-66}$, while none (0/4) of the USA-H isolates had either of these carbapenemases (Supplementary Fig. 9a). Interestingly, USA-H isolates clustered close together with most other lineage 7 PAK-H samples rather than as a separate group (Fig. 7a). A total of 92.3% (72/78) of the bacteria were resistant to three or more classes of antimicrobials including two carbapenems (Supplementary

Fig. 10a). A total of 4.05% (3/74) of the PAK-H *A. baumannii* isolates were resistant to all 14 antimicrobials tested. Minocycline was most efficacious against PAK-H strains, with 92.3% (72/78) non-resistant.

*E. faecium* isolates had 20 unique resistance genes against seven classes of antimicrobials (Fig. 7a). Only *erm(A)* was unique to a single isolate. Components of the *vanA* operon and the macrolide

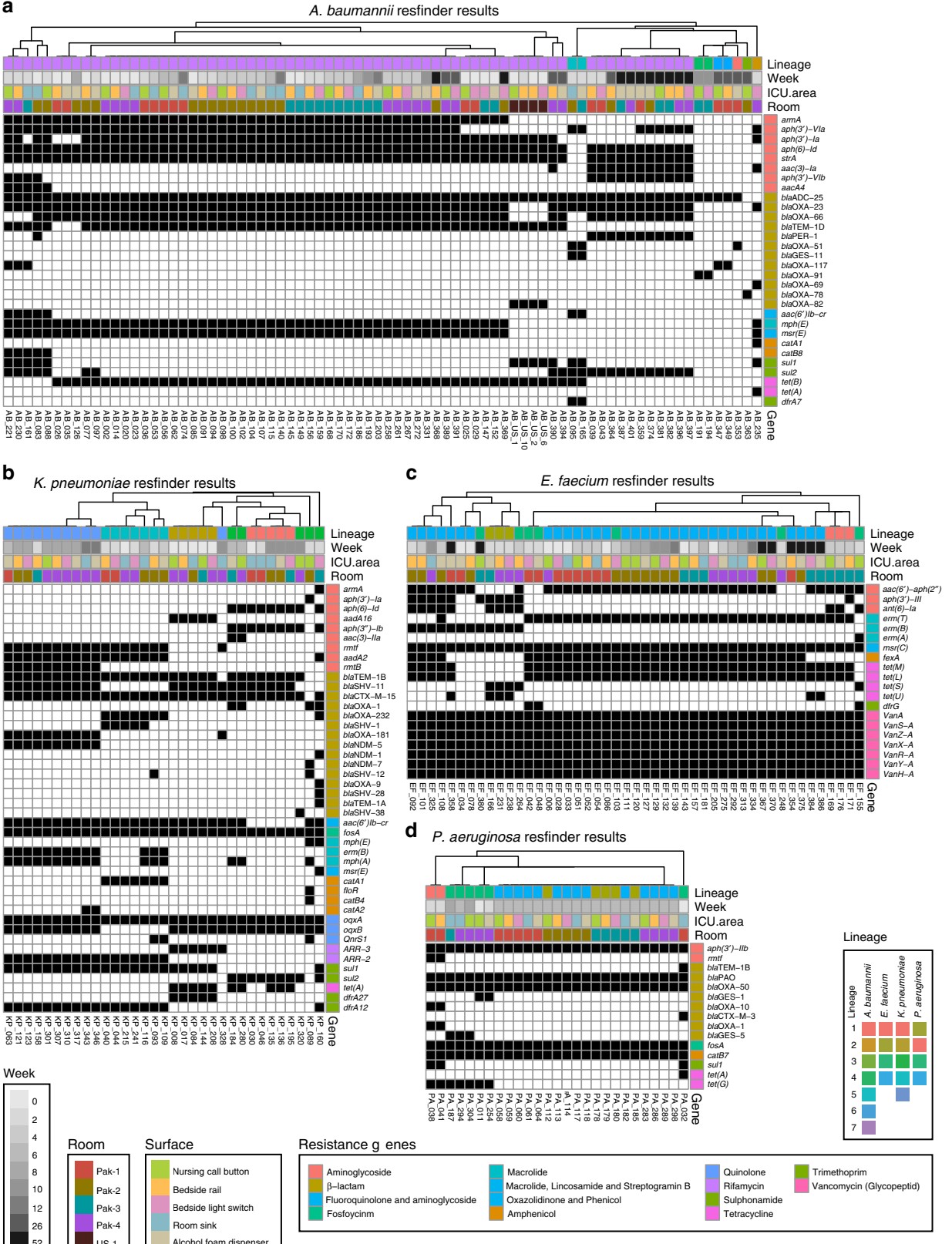

**Fig. 7** Genotypic antibiotic resistance in major species. Resfinder results for *A. baumannii* (**a**), *K. pneumoniae* (**b**), *E. faecium* (**c**), *P. aeruginosa* (**d**). Resistance genes are grouped by antibiotic class on the *y*-axis and individual isolates are hierarchically clustered by their resistance genes on the *x*-axis. Black squares indicate the presence of a specific resistance gene in an isolate. Colored annotations are added next to the resistance genes for resistance gene class and above the charts for hierBAPS lineage (identified in Fig. 3 by branch color), week, surface, and room. Source data for all panels are provided in the source data file

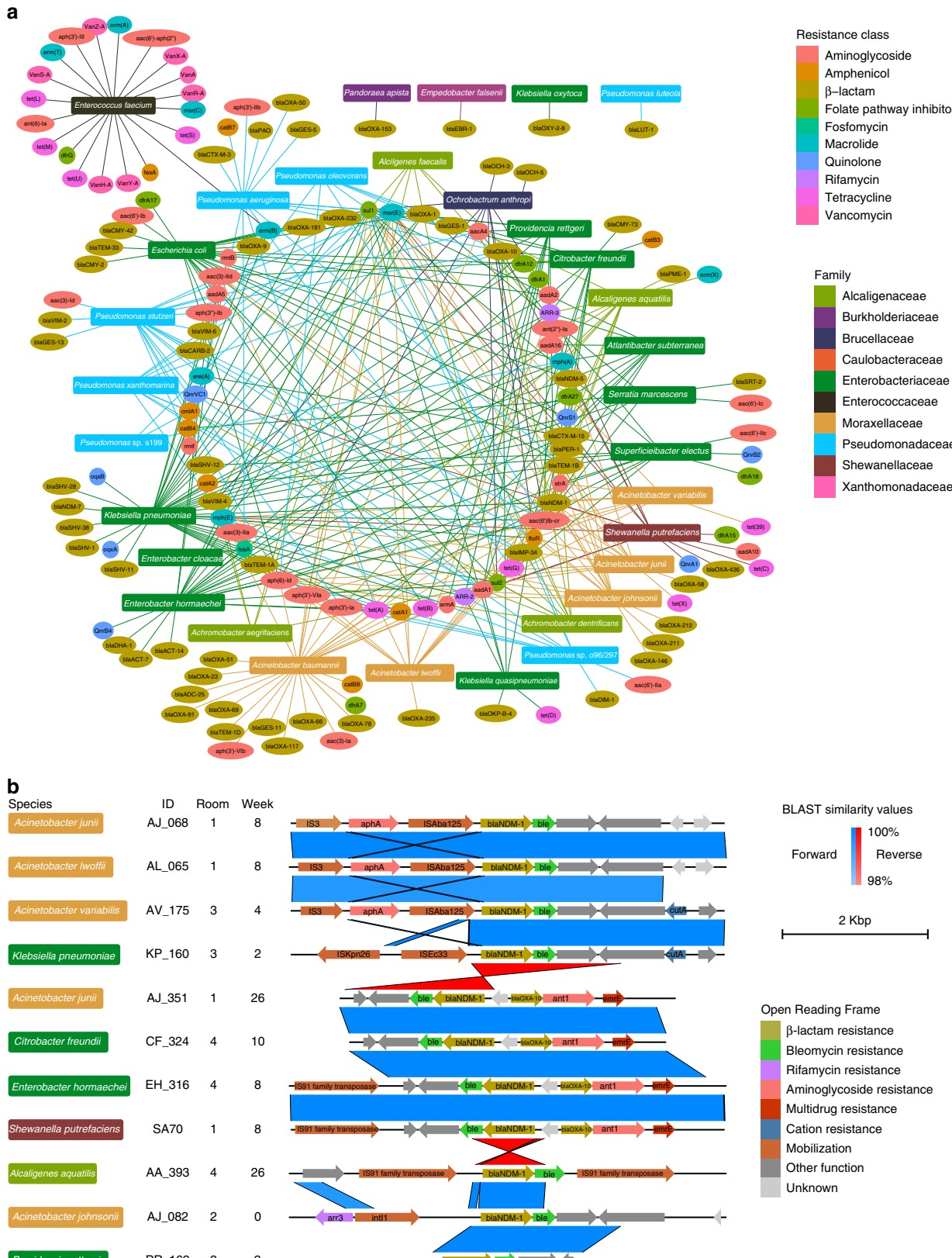

**Fig. 8** Shared antibiotic resistance genes across diverse taxonomic groups. **a** Species and resistance gene network diagram. Species are represented as rectangles colored by family. Resistance genes are represented by ovals colored by resistance gene class. Lines colored by species family are drawn from each species to all the resistance genes annotated by Resfinder in that species isolates. **b** Annotated *bla*NDM-1 contigs in 11 isolates. Protein annotations colored by putative function are shown as arrows for each isolate's *bla*NDM-1 contig. BLAST similarity values greater than 98% between contigs are shown in blue if they are oriented in the forward direction and red if they are oriented in the reverse direction. Species names are shown on the left in rectangular boxes colored by family and isolate ID, room, and week are also labeled. Source data for both panels are provided in the source data file

ARG *msr(C)* were common to all isolates. As expected for *vanA* containing *E. faecium*, all isolates were resistant to vancomycin (Supplementary Fig. 10b). A total of 42.1% (24/57) were additionally resistant to chloramphenicol and doxycycline. All isolates were susceptible to daptomycin.

The *K. pneumoniae* isolates harbored 44 unique resistance genes and of these, 25.0% (11/44) were unique to single isolates (Fig. 7c). Three *bla*$_{NDM}$ (*bla*$_{NDM-1}$, *bla*$_{NDM-5}$, and *bla*$_{NDM-7}$) and two *bla*$_{OXA}$ (*bla*$_{OXA-181}$ and *bla*$_{OXA-232}$) carbapenemase genes were identified. *bla*$_{NDM-5}$ was found in *K. pneumoniae* on ten surfaces and in all four PAK-H ICU rooms (Supplementary Fig. 9b). A total of 39.4% (13/33) of *K. pneumoniae* isolates were resistant to meropenem and imipenem (Supplementary Fig. 10c). One-hundred percent of lineage 1 (5/5) and lineage 2 (5/5) isolates and 60% (3/5) of lineage 3 isolates were susceptible to these two antibiotics. All (33/33) isolates harbored the fosfomycin ARG *fosA* and an efflux pump component *oqxA*, however all lineage 4 isolates lacked the second component, *oqxB*.

*P. aeruginosa* isolates harbored 15 unique resistance genes against six classes of antimicrobials. All isolates had *aph(3')-lb*, *bla*$_{PAO}$, *bla*$_{OXA-50}$, *fosA*, and *catB7* (Fig. 7d). Fifty percent (3/6) of lineage 3 genomes had the carbapenemase *bla*$_{GES-5}$. All lineage 4 *P. aeruginosa* isolates and 3/5 lineage 1 isolates were pan-susceptible to antibiotics (Supplementary Fig. 10d). In contrast, all (8/8) lineage 2 and 3 isolates were resistant to meropenem, ciprofloxacin, and gentamicin. Our results demonstrate that the major abundant HAI pathogens contain a high ARG burden and exhibit profound levels of multidrug resistance. Infections from these bacteria could have limited treatment options due to high phenotypic multidrug resistance.

**ARGs against almost all antimicrobials are shared between species**. Given the extensive diversity and burden of high-risk ARGs found in *A. baumannii*, *E. faecium*, *K. pneumoniae*, and *P. aeruginosa*, we analyzed potential lateral transfer of ARGs between all collected species. To accomplish this, we concatenated identified acquired ARGs within each species and created a network diagram connecting each taxa with its ARGs (Fig. 8a). The high connectivity of this network highlights the extensive promiscuity of ARGs we observe in these data. Strikingly, 57 ARGs were found in two or more species. These genes were expected to confer resistance against all classes of antibiotics, excluding vancomycin. *E. faecium* contained the macrolide resistance gene *erm(B)*, which was also shared with *E. coli*. Given that *E. faecium* is the sole gram-positive species in this collection, it unsurprisingly had the most species specific ARGs (*n* = 17). *Sul1* was the most promiscuous ARG within our cohort, as it was identified in 22 different species, including those in *Acinetobacter*, *Achromobacter*, *Alcaligenes*, *Atlanibacter*, *Citrobacter*, *Escherichia*, *Enterobacter*, *Klebsiella*, *Ochrobactrum*, *Pseudomonas*, *Providencia*, *Shewanella*, and *Superficieibacter*. β-lactam ARGs were the most abundant class in our cohort, with a total of 57 identified from all four Ambler classes. Alarmingly, 40.3% (23/57) of these genes have putative carbapenemase activity. *bla*$_{GES-5}$ is the only Ambler Class A carbapenemase. A total of 34.7% (8/23) of genes we identified are Ambler Class B Metallo-β-lactamases, from the *bla*$_{VIM}$, *bla*$_{IMP}$, *bla*$_{EBR}$, *bla*$_{DIM}$ and *bla*$_{NDM}$ families. The remaining 60.8% (14/23) were *bla*$_{OXA}$ variants including the *bla*$_{OXA-48-like}$ family members *bla*$_{OXA-181}$ and *bla*$_{OXA-232}$. *bla*$_{NDM-1}$ showed the greatest diversity of host species, as it was identified 11 times in ten different species from *Alcaligenaceae*, *Enterobacteriaceae*, *Moraxellaceae*, and *Shewanellaceae*.

*bla*$_{NDM}$ is a globally proliferated family of carbapenem resistance genes endemic to India and Pakistan[9,41]. To better understand the local genetic context of *bla*$_{NDM-1}$, we performed long-read sequencing with the Oxford NanoPore MinION platform on all *bla*$_{NDM-1}$ positive isolates (Fig. 8b). *bla*$_{NDM-1}$ in all genetic contexts was adjacent to *ble*, a bleomycin resistance gene. The *bla*$_{NDM-1}$ locus region was nearly identical between *A. junii* AJ_068/*A. lwoffii* AL_065/*A. variabilis* AV_175 and *A. junii* AJ_351/*C. freundii* CF_324, *E. hormaechei* EH_316, and *S. putrefaciens* SA70. The isolates *A. junii* AJ_351, *C. freundii* CF_324, *E. hormaechei* EH_316, and *S. putrefaciens* SA70 additionally contained *bla*$_{OXA-10}$ and *ant1*. The isolates *A. junii* AJ_068, *A. lwoffii* AL_065, and *A. variabilis* AV_175 had a different aminoglycoside resistance gene, *aph*. The isolate *A. johnsonii* AJ_082 contained the only rifamycin resistance gene, *arr3*. The isolates *A. junii* AJ_351, *C. freundii* CF_324, *E. hormaechei* EH_316, and *S. putrefaciens* SA70 also contained the *emrE* multidrug resistance transporter. On 72.7% (8/11) of the loci, *bla*$_{NDM-1}$ was co-localized with a transposase associated gene. Our analysis of ARG content across species identified high interconnectivity between most gram-negative species and determined *bla*$_{NDM-1}$ is situated in similar genetic contexts across diverse taxonomic groups, suggesting extensive horizontal ARG transfer.

**A. baumannii and E. faecium have synergistic biofilm interactions**. Bacteria harboring diverse ARGs may be recalcitrant to treatment regimens and could continually transmit from patients onto ICU surfaces, likely forming sessile biofilms to survive the dry conditions[42,43]. Indeed, biofilms composed of MDROs have been previously demonstrated to contaminate 93% (41/44) of hospital surfaces surveyed[24]. To assay potential microbe–microbe interactions that may explain long-term surface persistence, we examined co-occurrences between abundant species in the first three collection months using permutation testing (Supplementary Data 8). To remove potential bias from overrepresentation of certain taxa, we performed this analysis with both total counts (Supplementary Data 9) and relative frequency (Supplementary Data 10). Both metrics demonstrated *A. baumannii* and *E. faecium* co-occurred on surfaces more often than predicted by chance (*P* <0.00001 for *A. baumannii* and *P* = 0.0083 for *E. faecium* (permutation test)) (Fig. 9a, b).

We then obtained isogenic strains of *E. faecium* (TX82/TX5645) and *A. baumannii* (ATCC-17978, 17978 ΔpgI) capable of, or deficient in, biofilm formation, respectively[44,45]. Using every pairwise combination between the different species, we found co-culture of *E. faecium* TX82 with *A. baumannii* ATCC-17978 or *A. baumannii* 17978Δpgl, and *E. faecium* TX5645 with *A. baumannii* ATCC-17978 resulted in statistically significant increases (*P* < 0.0001 (Mann–Whitney *U* test)) in biofilm biomass relative to either of the parent strains (Fig. 9b). This effect did not occur when both species were incapable of forming biofilms individually (Fig. 9e).

As dead cells may be included in total analysis of biofilm biomass, we next specifically quantified the population of total viable cells between each pairwise interaction. Like results for total biofilm biomass, the number of viable cells increased significantly in *E. faecium* TX82/*A. baumannii* ATCC-17978 and *E. faecium* TX82/*A. baumannii* 17978Δpgl compared to either parent strain (*P* < 0.0001 (Mann–Whitney *U* test)) (Fig. 9d). However, in contrast to the increase in biofilm biomass observed for *E. faecium* TX5645/*A. baumannii* ATCC-17978 relative to both parent strains, we found a decrease in viable cells compared to *A. baumannii* ATCC-17978 (Fig. 9f). Quantification of biofilm biomass synergy values between each strain combination shows all interactions except those between *E. faecium* TX5645 and *A. baumannii* 17978Δpgl are synergistic. For viable cells, interactions between *E. faecium* TX5645 and *A. baumannii* 17978Δpgl

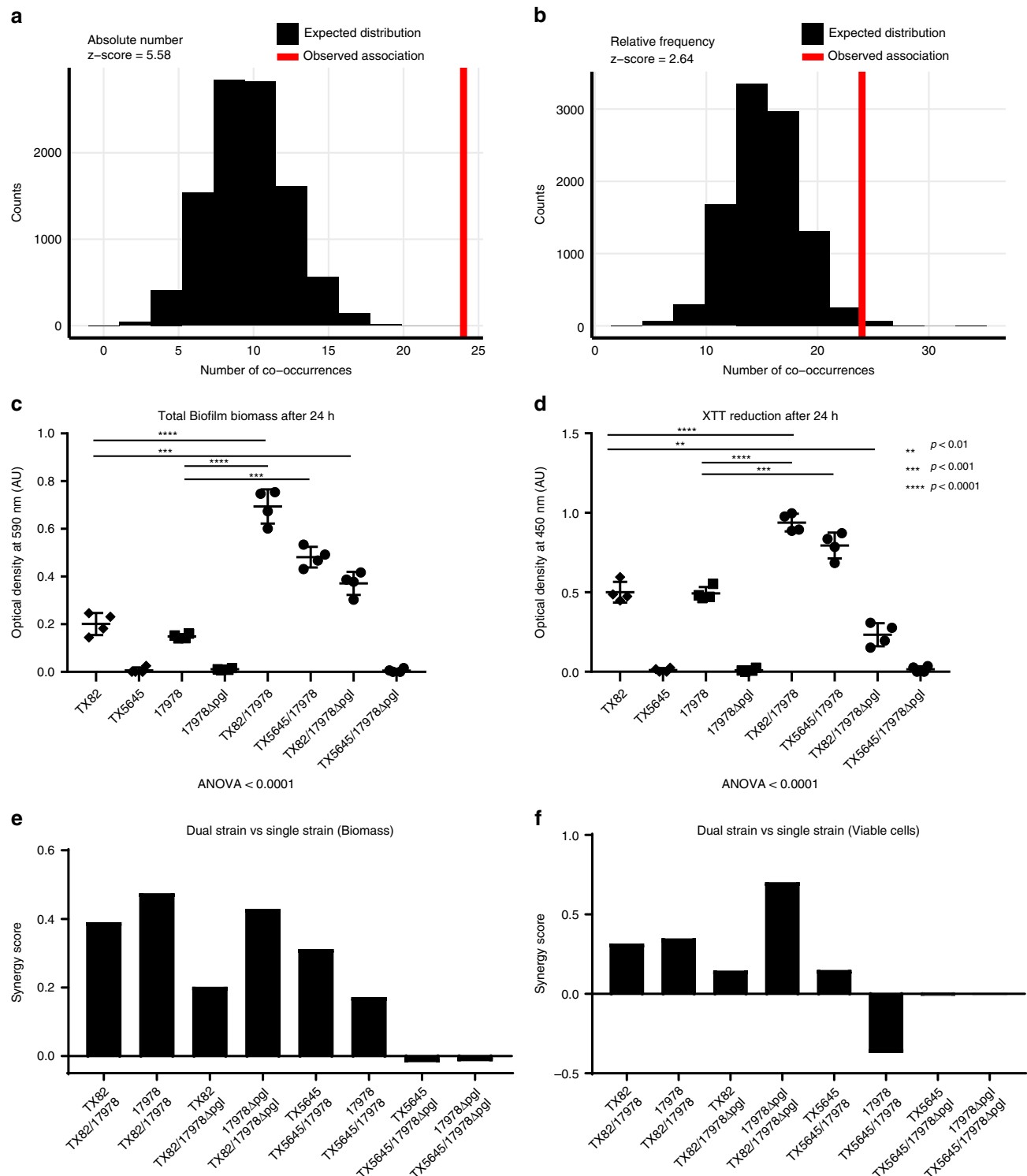

**Fig. 9** Synergistic biofilm interactions for *A. baumannii* and *E. faecium* predicted by surface collections. Permutation test of co-association between *A. baumannii* and *E. faecium* on surfaces conducted using species **a** absolute counts and **b** relative species frequencies. The expected distribution of the number of co-occurrences is shown in red and the observed number of co-occurrences in the dataset is shown as a vertical blue line. Total crystal violet stained **c** biofilm biomass and **d** XTT reduction for *A. baumannii* and *E. faecium* model biofilm strains grown in single and in co-culture (P-values were generated using unpaired, nonparametric Mann–Whitney statistical tests are indicated using the following mapping: **<0.01, ***<0.001, ****<0.0001). y-Axis for both plots is optical density at 590 nm and 450 nm, respectively, and error bars are 1 standard deviation. Synergy scores of dual vs single strain cultures for **e** biofilm biomass and **f** viable cells. Source data for all panels are provided in the source data file

and *A. baumannii* ATCC-17978 versus *E. faecium* TX5645/*A. baumannii* 17978Δpgl are synergistic. These data suggest interspecies interactions between organisms identified on PAK-H ICU surfaces may enable increased survival due to synergistic growth inside biofilms. Importantly, relative efficacy of those interspecies biofilms depends strongly on individual strain capabilities.

## Discussion

HAIs are a substantial patient health threat and economic burden[46]. While pathogenic bacteria that often cause HAIs can be transferred via invasive medical procedures or directly between patients or healthcare providers, inanimate surfaces and shared equipment are also an important reservoir for bacterial transmission[15,42]. Here we report an in-depth, year-long investigation of bacterial colonization of hospital surfaces in two ICUs in Pakistan (PAK-H) and the USA (USA-H). We found substantially more contamination by MDROs on PAK-H surfaces compared to USA-H surfaces using identical differential and selective culture conditions.

In addition to commonly recognized HAI causing bacteria, we found many potentially opportunistic pathogens and novel genomospecies from commonly pathogenic genera (*Pseudomonas*, *Stenotrophomonas*, *Brevundimonas*). The first novel genomospecies from this collection to be fully characterized, *S. electus*, is a new genus of *Enterobacteriaceae* that harbored extended spectrum β-lactamases and was multidrug resistant[31]. A previous taxonomic investigation determined that species from another novel genus of *Enterobacteriaceae*, *Pseudocitrobacter faecalis* and *Pseudocitrobacter anthropi*, harbored $bla_{NDM-1}$ carbapenemases, and were identified in fecal samples from patients at hospitals in Pakistan[47]. Currently no clinical evidence indicates these three species are human pathogens, but it is concerning that they exist proximal to known pathogens, encode clinically relevant ARGs, and are phenotypically resistant to multiple drugs. Furthermore, increasing implementation of WGS in clinical laboratories is enabling identification of emerging pathogens, which were previously misidentified by traditional methods, such as the first report of a bloodstream infection by *Kosakonia radicincitans*[48]. Our results provide additional utility for the implementation of WGS for bacterial delineation from clinically relevant environments. Further comparative analysis and molecular and phenotypic evidence for pathogenesis is required to demonstrate that this level of identification is clinically relevant or actionable.

*A. baumannii*, *E. faecium*, *K. pneumoniae*, and *P. aeruginosa*, the four most abundant bacteria in our cohort, are also common pathogens and common HAI agents. Interestingly, through core genome phylogenetic analysis, we found that our *A. baumannii* and *E. faecium* isolates are dominated by single lineages, but *K. pneumoniae* and *P. aeruginosa* have nearly equal numbers of isolates from multiple lineages. Previous reports of *K. pneumoniae* and *E. cloacae* isolates from a US hospital system and Italy showed they were similarly composed of diverse sequence types[49,50].

Timepoint of sample collection was the variable that showed greatest concordance with phylogenetic lineage. Lineage 7, the main group of *A. baumannii* isolates, was composed of several sequence types, including ST218, ST208, and ST195. These STs correspond to major strains collected of $bla_{OXA-23}$ bearing *A. baumannii* in Indonesia; additionally, $bla_{OXA-23}$ positive ST195 isolates were responsible for an outbreak of infections in North China[51,52]. The four ST208 USA-H *A. baumannii* isolates were genomically similar to the PAK-H isolates, although the PAK-H isolates harbored $bla_{OXA-23}$ whereas the USA-H isolates have $bla_{OXA-81}$. This parallels a previous investigation that whichfound

near identical genomes and plasmids from carbapenem-resistant *Enterobacteriaceae* in the US and Pakistan, but US isolates exclusively contained $bla_{KPC}$ while $bla_{NDM}$ was only found in isolates from Pakistan[30]. The most abundant *E. faecium* sequence type, ST132, was primarily contained in lineage 4. Isolates from this sequence type have been reported as both etiological agents of urinary tract infections and as commensal bacteria in animals[53,54].

Though *A. baumannii* and *E. faecium* were dominated by a single lineage, we found that the six clones of *E. faecium* came from all four lineages, but that 3/6 of the clones were from the dominant lineage 4 group. In contrast, all identified *A. baumannii* clones were in the dominant lineage 7. Given that clone 5 of *E. faecium* was found in 8/9 timepoints during our collections, including the first and last weeks of the year-long sampling period, it is possible that PAK-H surfaces are being colonized by a common seeding source or that these isolates represent the predominant clone circulating in the PAK-H region. Source investigation of carbapenemase producing organisms in a US hospital system determined that plasmids mobilizing the ARGs originated from building plumbing[55]. As further evidence of this, we found that *A. baumannii* and *E. faecium* clones are more likely to co-localize in space and time than if they were randomly distributed. This may have important clinical ramifications, as one analysis determined that although only 8.7% of ICU bacteria sequenced are from a clonal lineage, they were associated with clinical infection in 62% of occurrences[39]. Therefore, eradication of the common contaminating source could drastically reduce spread of these clones and thereby reduce potential of spread to hospital patients. If bacteria are transmitting between surfaces, spatial and temporal linkage of these surfaces could mean effective decontamination of surfaces will have a combinatorial effect.

In our variant analysis, we identified that most cliques (complete subgraphs within the network where each node in the clique is connected to each other node in the clique) were time restricted, but a few cliques persisted across multiple collections. These persistent cliques subsequently showed room restriction. Several contamination routes could explain these results. For example, seeding bacteria may originate from patients occupying hospital rooms[56]. Bacteria coming from different patients are likely genetically distinct in variant analysis; even within a patient, multiple lineages of the same species could co-exist[57]. Seeding events from patient to surface would represent a bottleneck event and persistence on surfaces would represent another bottleneck. Bacteria passing the first bottleneck would be detected within a single collection time, and bacteria passing the second would be found in multiple collection times. Bacterial clones on many surfaces would have higher chances to spread to other surfaces in the same room or different rooms. Similar contamination patterns could also be observed due to water contamination in the hospital[55]. PAK-H uses tap water with Virkon S disinfectant tablets (Lanxess) to clean hospital surfaces. If tap water has high bacteria burden or if not enough tablets are used, the disinfectant protocol could contaminate rather than decontaminate surfaces. This tap water environmental source could contain a polymicrobial community, thus acting as reservoir for multiple bacterial lineages[58]. With tap water, the first significant bottleneck would be getting from the water system to surfaces, but subsequent steps would be in line with the patient contamination scenario. In support of these potential contamination routes, the bacteria we observe in this study are a mixture of human fecal bacteria and water environmental bacteria[59,60]. This analysis demonstrates how a surface focused sampling and analysis approach can generate epidemiologically meaningful insights for future investigation. In our case, the hospital water system and ICU room patients can both be tested as potential reservoirs for

observed ICU surface bacterial contaminants, and a longitudinal sampling scheme similar to the one used in our study would enable estimation of transmission dynamics between these putative contamination sources and sinks.

The *A. baumannii*, *E. faecium*, *K. pneumoniae*, and *P. aeruginosa* isolates we recovered from PAK-H surfaces had high ARG burdens and were often phenotypically resistant to multiple classes of antibiotics commonly used as treatment against them. This is particularly troublesome for local patient safety at PAK-H given that a retrospective cohort analysis found significant increases in 30-day mortality after infection when comparing patients infected by multidrug-resistant versus susceptible organisms[61]. Particularly problematic are the three *A. baumannii* isolates we recovered that were resistant to all antibiotics tested with CLSI interpretive criteria, similar to 20 pandrug-resistant isolates recovered from countries bordering the Mediterranean Sea[62]. While we were unable to determine directionality of transfer, linkage analysis between acquired ARGs and species harboring them show numerous instances of identical ARGs in different species. This is best exemplified by $bla_{NDM-1}$ presence in ten different species. Using long-read nanopore sequencing, we found $bla_{NDM-1}$ situated in a variety of genetic contexts, even between the two *A. junii* isolates that contained it. Similar to previous reports of $bla_{NDM-1}$ in isolates recovered from Pakistan patient stool samples, the mobilization element ISAba125 was co-localized with $bla_{NDM-1}$ in 4/11 of our isolates[63]. Additionally, 4/11 isolates also contained $bla_{NDM-1}$ close to $bla_{OXA-10}$, similar to numerous $bla_{NDM-1}$ harboring Enterobacteriaceae isolates from hospitalized patients[63].

Bacteria surviving in the built environment likely exist in sessile biofilms, which can make them difficult to eradicate[58]. Numerous studies have determined that dual- or multi-species biofilms have distinct characteristics to enhance survival and pathogenicity[64–66]. Direct sampling of ICU samples showed polymicrobial biofilms are widespread[24]. Biofilm formation is an important component for pathogenesis of *Enterococcus* and *Acinetobacter*[67,68]. In both organisms, biofilm formation often requires extracellular attaching proteins including LH92_11085 and OmpA in *A. baumannii* or the Emp pilus in *E. faecium*[69–71]. Variation has been observed among the ability of *A. baumannii* clinical isolates to form biofilms, but several strains are capable of growing on urinary catheter surfaces[72]. In *E. faecium*, adaption to a biofilm is associated with changes in the transcriptional program[73]. 16S rRNA gene sequencing of high-touch surfaces at large public hospitals in Brazil identified both *E. faecium* and *A. baumannii* co-localized to the same surface[74]. Despite this observation and the role of individual genes in biofilm formation for both species, there is a dearth of relative knowledge on specific interactions between these two species that may occur in the built environment. Our analysis of co-occurrence between organisms indicates *A. baumannii* and *E. faecium* isolates were cultured together more frequently than expected by chance. Additionally, we found co-culture of model *E. faecium* and *A. baumannii* biofilm-forming and biofilm-deficient strains resulted in changes in total biofilm biomass and total viable cells dependent on the biofilm formation capacity of input strains. These results are consistent with a previous report on changes between *Enterococcus faecalis* and *P. aeruginosa* biofilms, where synergistic interactions between the exopolysaccharide produced by *P. aeruginosa* is responsible for spatial segregation of the two species in biofilms[75]. It is therefore possible that conserved interspecies interactions between *Enterococcus* spp. and gram-negative non-fermenting bacteria may explain prolonged surface survival.

One limitation of our study is some bacterial species may be more robust than others in surviving on surfaces and in the sampling protocol. For example, bacteria could exist transiently between sampling times in concert on surfaces. However, the number of rare species we collected helps to allay this concern. We also did not concurrently characterize isolates recovered from clinical specimens. Therefore, we are unable to determine if lineages found on surfaces correlate with lineages associated with clinical infection in the hospital and in addition, we cannot corroborate linkage of lineages (e.g. *P. aeruginosa* in week 4) or clones (e.g. *E. faecium* clone 5) with time to determine if outbreaks occurred. Detailed analysis of temporally matched clinical isolates may additionally inform associations of identified *A. baumannii*, *E. faecium*, *K. pneumoniae*, or *P. aeruginosa* lineages with specific infection niches and elucidate novel virulence factors or identify contaminated medical equipment. Additionally, synchronous sampling of patient/healthcare workers' skin, stool, or oral microbiota, or of the room plumbing system could be used to further track transmission of the recovered MDROs to a specific source.

Our work represents a thorough longitudinal analysis of hospital surface contamination in Pakistan. We unequivocally demonstrate that MDRO burden is higher on PAK-H surfaces than on analogous USA-H surfaces. Using WGS we found that while the recognized human pathogens *A. baumannii*, *E. faecium*, *K. pneumoniae*, and *P. aeruginosa* are the most abundant organisms, a variety of potentially pathogenic taxa and novel genomospecies were also recovered. Analysis of lineages in the four most abundant species and clones in *A. baumannii* and *E. faecium* provide evidence of a common point source of contamination. Particularly alarming is our determination that these isolates harbor a high burden of ARGs, are often phenotypically multidrug resistant, and that identical ARGs are housed on a variety of genetic platforms in multiple species. Synergistic growth of *E. faecium* and *A. baumannii* in dual-species biofilms may explain statistically significant co-occurrence on PAK-H surfaces. The complex built-environment microbial ecology revealed by our hospital sampling highlights that common human pathogens and rare species frequently colocalize and share clinically relevant genes. Rapid dissemination of bacterial pathogens and plasmid borne ARGs stress the importance of surveilling bacterial isolates in high-risk areas to protect vulnerable hospitalized patients around the globe.

## Methods

**Sample collection and culturing**. ICU rooms were sampled every other week for 3 months and then at 6 months and 1 year after the initial sampling. At each time point, five surfaces were sampled in each patient room (if available in that room): the nursing call button (sampled the call button that is attached to the right of the bedside rail, swabbing as much of the surface as possible), the bedside rail (swabbing approximately 6 inches of the rail, swabbing the side that is closest to the room door), the main room light switch (swabbing the entire switch and switch plate), the sink handles (swabbing the handles on the sink inside the patient room, swabbing both handles, front and back), the alcohol hand foam dispenser (swabbing the one closest to the patient room, swabbing the high touch area of the dispenser). If a bedpan, commode or toilet was present in the patient room, this was also sampled, including the seat and handle. The Eswab collection and transport system (Copan, Murieta, CA) was used to collect all specimens; swabs were moistened prior to sample collection. Two swabs were held together for specimen collection. Specimens collected in Pakistan were shipped to the US site for workup and analysis.

One Eswab specimen was vortexed and 90 µL of eluate was inoculated to each of the following culture medium: Sheep's blood agar (Hardy Diagnostics), MacConkey agar (Hardy Diagnostics), VRE chromID (bioMerieux), Spectra MRSA (Remel), HardyCHROM ESBL (Hardy), Pseudo agar (Hardy), and MacConkey agar with cefotaxime (Hardy). Plates were incubated at 35 °C in an air incubator and incubated up to 48 h prior to discard if no growth. Up to four colonies of each colony morphotype (as appropriate for the agar type) were subcultured and identified using matrix-assisted laser desorption/ionization time-of-flight mass spectrometry (MALDI-TOF MS) with the VITEK MS system[76–80]. A second Eswab specimen was used for *Clostridioides difficile* culture with a heat-shock broth enrichment method as previously described[81]. All isolates recovered were stored at −80 °C in TSB with glycerol.

**Antibiotic susceptibility testing**. Antimicrobial susceptibility testing was performed using Kirby Bauer disk diffusion, interpreted according to CLSI standards[38].

**Illumina WGS**. Unique colony morphotypes from the initial swab plates were streaked for isolation on blood agar. After a culture was deemed pure by visual determination, ~10 colonies were suspended in deionized water with a sterile cotton swab. Total genomic DNA was extracted from the suspension using the bacteremia kit (Qiagen, Germantown, MD, USA). DNA was quantified with the Quant-iT PicoGreen dsDNA assay (Thermo Fisher Scientific, Waltham, MA, USA). A total of 5 ng/uL of DNA was used as input for Illumina sequencing libraries with the nextera kit (Illumina, San Diego, CA, USA)[82]. The libraries were pooled and sequenced on a NextSeq HighOutput platform (Illumina) to obtain 2 × 150 bp reads. The reads were demultiplexed by barcode, had adapters removed with Trimmomatic v.36, and contaminating sequences with Deconseq v.4.3[83,84]. Processed reads were assembled into draft genomes using the de-novo assembler SPAdes v3.11.0[85]. The scaffolds.fasta files were used for all downstream analysis. Assembly statistics on the assemblies was quantified using QUAST v4.5[86]. Prokka v1.12 was run on the scaffolds file to identify open reading frames >500 bp in length[33].

For the 11 isolates chosen to be sequenced with Nanopore technology, Genomic DNA was extracted using the Genomic-Tip 500/G (Qiagen) and genomic DNA buffer set (Qiagen) per manufactures instructions. The DNA was converted into a sequencing library on with the Rapid Barcoding Kit (Nanopore, Cambridge, MA, USA) per manufactures instructions and sequenced on the MinION platform. The output fastq files were used in a hybrid assembly with SPAdes v3.11.0 and processed Illumina reads.

These assemblies are uploaded to NCBI under BioProject PRJNA497126 (https://www.ncbi.nlm.nih.gov/bioproject/497126).

**Taxonomic assignment**. All isolates were initially identified using the VITEK MS MALDI-TOF MS v2.3.3. Following draft genome assembly, the species determination for all isolates were then investigated using an in silico approach. MASH was performed against all of the isolate genomes[87]. Isolates that had 100% concordance between the MALDI-TOF MS assignment and the top 10 MASH hits were determined to be the species assigned by MALDI-TOF MS. Isolates that had discrepant analysis were then manually investigated further, by using RNAmmer v1.2 to identify the 16S rRNA sequence, submission of that sequence to the EZ BIoCloud taxonomic database, and finally ANI analysis with the mummer method between the isolate in question and the appropriate type genome (if available) using the JSpecies webserver (http://jspecies.ribohost.com/jspeciesws)[88–90]. Species were determined if the genome in question had >95% ANIm with the type genome (if available), or >99% 16S rRNA identity (if type genome is not available)[91,92]. Isolates that did not pass either of these thresholds are therefore considered to be novel genomospecies. Finally, all the isolates sequenced in this study were used to construct a Hadamard matrix, representing the product of the ANI and percent genome aligned, with the ANIm method from pyANI (https://github.com/widdowquinn/pyani). The matrix was visualized using the python package Seaborn (http://seaborn.pydata.org) and annotated for initial MALDI-TOF MS identification, and in silico assignment if discrepancies were identified.

**Core genome alignment**. The gff files produced from Prokka for *A. baumannii*, *E. faecium*, *K. pneumoniae*, and *P. aeruginosa* were used to construct a core genome alignment with Roary v3.8.0 and PRANK v1.0[34,93]. fastGEAR was ran on the respective core_genome_alignment.aln output of Roary to identify instances of recombination within these species[36]. The recombinant regions were removed using custom python scripts. The recombination purged core genome alignment was used to generate a maximum likelihood tree with RAxML v8.2.11[35]. The output newick file was visualized in iTOL. In silico multilocus sequence typing (MLST) was performed with the MLST program. The sequence type information, week of collection, room of collection, and surface was viewed as a color strip in iTOL[94]. Lineages identified by hierBAPS during fastGEAR were also marked on the trees[95].

**Clonality analysis**. Pairwise SNP counts between all isolates in the recombinant corrected core genome alignment were calculated. All paired distances >5 SNPs were excluded from further clonality analysis. Pairwise groupings with five or fewer SNPs were imported to Gephi as an unweighted pairwise links table. Gephi's built in modularity analysis was used to isolate perfectly reciprocal groupings. R was used to visualize these groupings in Fig. 4.

Pairwise SNP distances were calculated as the number of SNPs between two isolates divided by the total number of positions in the core genome alignment.

**Linking *A. baumannii* and *E. faecium* clones to accessory genomes**. Principal coordinates analysis was done using a gene presence or absence matrix from Roary. Core genes were removed from the matrix and the vegdist function from the Vegan package in R and pcoa function from the ape package in R were used to compute the distance matrix and principal coordinate decomposition respectively for each bacteria. Centers of gravity for each clonal group and for the non-clonal bacteria

were calculated. This entire process was repeated without the non-clonal bacteria. R was used to visualize the first and second axis for each principal coordinate decomposition.

**Spatiotemporal clone linkage analysis**. Pairwise spatial and temporal distances for all surfaces in the first 3 months of collections were calculated. For spatial distance, the pairwise distance of the same surface from different times of collection was given a distance of 0. Pairwise distances of surfaces in the same room were given distances of +1 and pairwise distances of surfaces from different rooms were given distances of +2. For temporal distance, each 2-week span was counted as a distance of +1. Thus, surfaces 4 weeks apart in collection time were given a distance of 2. Spatiotemporal linkage analysis was conducted for clones present in the first 3 months of collection with more than one isolate. Observed distances were calculated by adding together the spatial or temporal distances for that clone using their observed collection locations and times. Expected distributions were calculated by conducting 10,000 permutations of the spatial and temporal distances in the dataset and then logging the permuted distances for each clone. For Fig. 4c, d observed distances for all clones were summed to get the observed values and all clones were summed by permutation number to get the expected distance distributions. R was used to visualize these observed values and permuted distributions.

**Variant calling**. Snippy v4.3.8 (https://github.com/tseemann/snippy) was used to map forward and reverse reads for each isolate to the type strain complete genome assembly (GCF_000746645.1 (https://www.ncbi.nlm.nih.gov/assembly/GCF_000746645.1/) for *A. baumannii* and GCF_000174395.2 (https://www.ncbi.nlm.nih.gov/assembly/GCF_000174395.2/) for *E. faecium*) and to call variants. Resultant VCF variant files were merged using BCFtools v1.9 with *-m all* option for multiallelic records. Merged VCF files were parsed using vcfR[96] and custom python scripts to get pairwise variant distances between all isolates. See Supplementary Fig. 6 for a brief overview of this process.

**Isolate clique grouping by pairwise variant distance**. Isolates were grouped into cliques at every pairwise distance cutoff found between isolates starting from two most similar isolates and ending at the most similar out of lineage comparison. At each distance cutoff, the variant pairwise distance matrix was filtered to only include comparisons below the cutoff threshold. The pairwise distance matrix was converted to a weighted edgelist. Weights were assigned using Eq. 1.

$$1 - (variant\_distance/max(variant\_distance)). \tag{1}$$

This weighted edgelist was used to make a undirected graph using the *graph_from_data_frame* function in igraph v1.2.4.1. Perfectly reciprocal cliques of maximal graph coverage were then identified using a greedy approach. The *maximal.cliques* function was used to identify perfectly reciprocal (fully connected) cliques. The identified cliques were then sorted by the highest minimum edge weight to get clique with strongest internal connectivity. The nodes from this "strongest" clique were then removed from the edgelist and the algorithm was run recursively until all possible fully connected cliques with two or more nodes were identified. See Supplementary Fig. 8 for a brief overview of this process.

**Calculate temporal and spatial distances for variant cliques**. Spatial and temporal analysis for variant cliques used the same distances as core genome SNP linkage analysis. Spatiotemporal linkage analysis was conducted for isolates in the first 3 months of collection. For each cutoff value, observed distances were calculated by adding together the spatial or temporal distances within clique and expected distributions were calculated by conducting 10,000 permutations of the spatial and temporal distances using the *sample* function in R v3.53. Thus, permutations kept clique structure, but shuffled distance information.

**ARG identification**. Acquired ARGs against aminoglycosides, amphenicols, β-lactam, folate pathway inhibitors, fosfomycin, macrolides/lincosamides/streptogramins, quinolones, rifamycin, tetracycline, vancomycin were annotated using the ResFinder BLAST identification program[37]. For the abundant species, the presence/absence matrix of ARGs was visualized in pheatmap (R). Associated metadata was displayed as a color strip to represent bacterial isolate demographics and expected resistance to antibiotics. To identify connectivity between the recovered species from the Pakistan ICU, we constructed a Source/Target/Edge formatted file, where each source represented a novel or curated genomospecies, a target was the unique ARG, and Edge weight was determined to be the number of times that ARG was identified within that species. The file was visualized in Cytoscape v3.4.0[97].

***bla*NDM-1 loci annotation and comparison**. A ~6–2 kB series of nucleotides flanking the *bla*NDM-1 loci in all positive strains was manually retrieved from SPAdes output of MinION & Illumina hybrid assembly. The nucleotides were re-annotated with prokka. The .gff file was used as input for Roary, to identify identical genes within the loci pan-genome. The .gbk files from prokka were viewed for open reading frames and BLAST similarity in EasyFig[98]. The sequences were ordered by their relationship from the newick tree created from the presence/

absence matrix of genes. All loci in the pan-genome were submitted to BLASTX against the refseq_proteins in October 2017 to identify a putative function[99]. The pairwise BLAST similarity was visualized on the EasyFig v2.2.2 construction by BLASTn similarity between the fasta files.

**A. baumannii and E. faecium co-association permutation testing**. Testing for significant association of A. baumannii and E. faecium was conducted using MALDI-TOF MS identifications from the first 3 months of PAK-H collections. The number and type of unique bacteria on each surface was tabulated. The number of surfaces with both A. baumannii and E. faecium was recorded as the observed frequency of co-occurrence. Absolute number and relative frequency expected distributions for A. baumannii and E. faecium co-occurrence were calculated using permutation tests with 10,000 random subsamples. For absolute number, the exact number of each bacterial species we collected was randomly distributed to a blank surface space with the restriction that each surface could not have more than one of the same species and that each surface had to get the same number of bacteria that was originally collected from the surface. This resulted a new permuted collection space with the same overall number of each bacterial species, but with randomized placement for each bacterium. The co-occurrence of A. baumannii and E. faecium for this permuted collection space was then recorded for the expected distribution. For relative frequency, the number of each species collected was used to calculate the frequency of that bacterial species in the collections. During permutation species were randomly chosen, weighted by their frequency in the collections. R was used to visualize the A. baumannii and E. faecium co-association expected distributions and observed values.

**Biofilm assays**. Frozen cultures of A. baumannii ATCC-17978 (17978), A. baumannii ATCC-17978Δpgl, E. faecium TX82, and E. faecium TX5645 were streaked onto tryptic soy agar (Difco, Detroit, MI, USA) and grown overnight at 37 °C. Isolated colonies were suspended in tryptic soy broth (Difco, Detroit, MI, USA) supplemented with 0.5% glucose (MP Biomedicals, Santa Ana, CA, USA) to promote the growth of E. faecium biofilm and quantified for OD600 using a 1:10 dilution. In concordance with previous investigations using respective strains, the A. baumannii isolates were normalized to 0.05 OD600 and the E. faecium were normalized to 0.10 OD600, for functional assays.

To grow biofilms, 200 μl of each single strain or 100 μl of A. baumannii and 100 μl of E. faecium dual species biofilms were added to tissue-culture-treated 96-well polystyrene microtiter plates (Sigma Aldrich, St. Louis, MO, USA) in triplicate. We additionally plated cell-free controls to ensure that no contamination occurred and to subtract out background absorbance reading. After pipetting, the plates were gently pipetted up and down to ensure that the strains mixed thoroughly. The plates were covered with breath ez membrane (Diversified Biotech, Dedham, MA, USA) and grown on the benchtop at ~22 °C for 16 h.

Following a growth period, the biofilm plates had planktonic cells removed by washing thoroughly with 250 μl of sterile phosphate buffered saline (PBS) (Thermo Fisher Scientific, Waltham, MA, USA) three times. To obtain the total biofilm biomass, the washed biofilms were fixed with 250 μl of bouin's solution (Sigma Aldrich) at 22 °C on the benchtop for 30 min. The fixative was washed three times with 200 μl of sterile PBS and then stained with 250 μl of 0.01% crystal violet (Sigma Aldrich) in water for 30 min at 22 °C on the bench. Finally, the unstained crystal violet was removed by washing three times with PBS and then the biomass was solubilized with 250 μl of 100% ethanol (Sigma Aldrich). The amount of biofilm biomass was quantified using nanometers absorbance with a Synergy H1 (BioTek) spectrophotometry machine. All raw absorbance values were adjusted by removing the background values obtained from the cell-free TSB controls. The conditions had average and standard deviation calculated.

For quantification of total viable cells in the biofilm, the biofilms were formed as previously described. After 16 h growth at 22 °C, planktonic cells were removed by washing thoroughly with 250 μl of PBS. The XTT cell viability kit (Cell Signaling Technologies, Danvers, MA, USA) was then performed according to manufacturer's instructions. The plates were read in the Synergy H1 spectrophotometry machine after 5-h incubation in the dark.

For the crystal violet and XTT reduction assays, the biofilm synergy scores were calculated as previously reported for dual-species biofilms. For each pairwise comparison, the synergy scores were reported as the difference between the average plus standard deviation for the single species biofilm and average minus standard deviation of the dual species biofilm.

$$\text{Biofilm synergy} = \left( \text{Average}_{\text{DualSpecies}} - \text{Standard Deviation}_{\text{DualSpecies}} \right)$$
$$- \left( \text{Average}_{\text{SingleSpecies}} + \text{Standard Deviation}_{\text{SingleSpecies}} \right). \quad (2)$$

**Statistics**. Unpaired, nonparametric Mann–Whitney statistical tests were used to compare the adjusted $OD_{590}$ and $OD_{450}$ values between the total biofilm biomass and total viable cells in the dual versus single species biofilms.

**Reporting summary**. Further information on research design is available in the Nature Research Reporting Summary linked to this article.

## Data availability

Assemblies and sequencing reads are available from NCBI under BioProject accession code PRJNA497126 (https://www.ncbi.nlm.nih.gov/bioproject/497126). Source data for all data figures are included as a Source Data zip file with additional subfolders by figure.

## Code availability

Example code for variant clique grouping depicted in Supplementary Fig. 8 is included as Supplementary Software.

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

## Acknowledgements

This work was supported by a United States Agency for International Development award (award number 3220-29047) to S.A., C.A.B., and G.D. This work was also supported in part by awards to G.D. through the National Institute of Allergy and Infectious Diseases and the Eunice Kennedy Shriver National Institute of Child Health & Human Development of the National Institutes of Health (NIH) under award numbers R01AI123394 and R01HD092414, respectively. JHK received support from the Washington University Institute of Clinical and Translational Sciences grant UL1TR000448, sub-award KL2TR000450 from the National Center for Advancing Translational Sciences of the NIH. R.F.P. received support from the Monsanto Excellence Fund Graduate Fellowship. A.W.D. received support from the Institutional Program Unifying Population and Laboratory-Based Sciences Burroughs Welcome Fund grant to Washington University. The content is solely the responsibility of the authors and does not necessarily represent the official views of the funding agencies. We thank Mario Feldman and his laboratory for providing strains of *A. baumannii* ATCC-17978 (17978) and *A. baumannii* ATCC-17978Δpgl. Additionally, we thank Barbara Murray and her laboratory for providing strains *E. faecium* TX82 and TX5645. The authors thank the Edison Family Center for Genome Sciences & Systems Biology at Washington University in St. Louis School of Medicine staff, Eric Martin, Brian Koebbe, and Jessica Hoisington-López for technical support in high-throughput sequencing and computing.

## Author contributions

A.W.D., R.F.P., J.H.K., S.A., C.A.B and G.D. conceived the study design, experiments, and analyses. M.W., A.S. and D.G. oversaw collection of samples and clinical metadata. A.W.D., R.F.P., M.W., A.S., S.P. and X.S. performed wet-lab experiments with advice from C.A.B. and G.D. A.W.D. and R.F.P. performed computational analyses with advice from C.A.B. and G.D. Article drafting was performed by A.W.D. and R.F.P. with critical revision performed by J.H.K., S.A., C.A.B. and G.D.

## Competing interests

The authors declare no competing interests.
