## [Peer Review File · Nature Communications]

Reviewers' comments:

Reviewer #1 (Remarks to the Author):

The manuscript "Transmission dynamics of multidrug resistant bacteria on intensive care unit surfaces" describes cultivable contaminants on ICU surfaces in selected hospitals in two countries with substantially different infection control standards. Contaminating microbes on surfaces in US hospitals have been well studied, and little new was found here. Contaminating microbes on surfaces of hospitals in Pakistan were much more abundant and diverse, which was largely predictable, but the specific identification of these microbes and the appearance and disappearance of specific lineages among them is useful for future development of infection control measures that are optimized for this environment. The genetic analysis of the microbes recovered goes farther than most other reports, and is done in a complete and competent manner, providing insights into the ebb and flow of microbes mainly in the Pakistan hospital environment, and a complete inventory of the antibiotic resistances that they harbor. Although the broad distribution of antibiotic resistances among various species is known, this study adds new information on their distribution among the organisms isolated. The study observed the co-occurrence of *E. faecium* with *A. baumannii*, and explored the ability of the two microbes to form interspecies biofilms. These were found to be robust leading to greater growth and persistence, supporting the prospect that a biological synergy may account for their co-occurrence on contaminated surfaces. While interesting, the studies were somewhat superficial and did not identify anything particularly proprietary about these interactions, or identify such interactions within biofilms recovered from hospital surfaces. Alternatively, it seems plausible if not likely, that as some of the most environmentally robust microbes, *A. baumannii* and *E. faecium* may co-occur among the last cultivable microbes on a surface because of their ability to survive as opposed to any particular biological synergy.

In summary, this is an unusually solid study of the microbes that contaminate hospital surfaces mainly in Pakistan. While robust with data of unusual depth, the findings of this report, in this reviewer's estimation, are largely predictable. As new resistances are emerging and spreading in developing economies where hygiene is difficult and antibiotics are abundant, this data however does provide a nice snapshot of the current situation.

The manuscript would benefit from more sharply defining the rationale for the study design, and from making more obvious to the reader what of the many bioinformatic analyses yielded results that break substantial new ground.

Reviewer #2 (Remarks to the Author):

In this work, D'Souza et al study environmental contamination in ICUs in a US hospital and a Pakistani hospital. The authors performed 180 samplings per hospital over the course of a year, followed by bacterial culture and identification of isolates. 295 Isolates were chosen for phenotypic characterization and genomic sequencing. The authors focus on *Acinetobacter*, *Enterococcus*, *Klebsiella* and *Pseudomonas* for further genomic analysis. The main findings include a remarkable level of contamination of surfaces in the Pakistani hospital, and the persistence of clones of *Acinetobacter* and *Enterococcus* on these contaminated surfaces. The authors explore a synergistic biofilm hypothesis to explain why both *A. baumannii* and *E. faecium* are found together on dirty surfaces.

Major Comments:

In this work, the authors performed a number of fairly standard analyses on the genomes of the large set of isolate collected, including phylogenetic comparisons and annotation of resistance genes. This manuscript represents clearly a substantial effort for which the authors are to be applauded. The analysis appears to be technically sound (though see below regarding genome assembly statistics). However, I believe the authors need to make a much better case for what is novel in their comprehensive data set. The finding of pervasive poor sanitation in the sampled Pakistani ICU is indeed alarming and actionable, but does not itself constitute an advance in biological knowledge. Nor does the finding that clonal strains of *A. baumannii* and *E. faecium* persist together on the contaminated environmental surfaces over time. These two high impact nosocomial pathogens have been studied in numerous works over the past two decades. Though I find the *E. faecium* and *A. baumannii* biofilm experiments very interesting, I think they should be better set in the context of the extensive work that has been done on biofilm formation by these two organisms. The title refers to "transmission dynamics". I think a good case is certainly made that dirty surfaces contain mixes of nosocomial clones that are clustered in space and persist over time, but I'm not sure I see the link with what would be called "transmission dynamics" per se, given that mechanism of movement of isolates between surfaces was not investigated, and patient samples were not included. (I do not believe this detracts from the quality of the study; it is a matter of how the scope is accurately conveyed by the title.)

The methods state that Illumina assembly statistics were quantified with Quast, but I can't find them in the manuscript. Standard assembly quality parameters must be provided for both the Illumina and

Nanopore assemblies. This should include genome assembly lengths, N50, number of contigs, mean + SD coverage statistics, etc. This can be done as a supplementary table.

Line 566 Clonality analysis. Why were the SNP counts for clonality made on Roary recombination-purged core genomes as opposed to whole genome alignments. The methods used by the authors can underestimate actual differences between isolates where deletions, insertions, duplications, etc. have occurred. Additionally and importantly, were indels (both small and large) included in any of the derivative analyses? Only SNPs are mentioned.

Minor Comments:

Figure 1A. Given the substantial differences between PAK-H and USA-H, the authors may consider representing this as two organism abundance histograms rather than one.

Figure 2. I find the organization of the legend with indented items to be confusing, and many of the indented items (disagreements between MALDI and the ANI-based identifications) appear to be missing or are not visible in the figure to which the legend refers - for instance *Acinetobacter soli* (1), *Pseudomonas sp. s199* (1), unless I am misinterpreting the figure. I would consider placing this figure in the supplementary data. Minor comment: Axis should be axes in the text.

Figure 3. It would be helpful in the figure text to indicate that the branch color coding in the phylogenetic tree in this figure corresponds to what the authors call "lineages" in Figure 4A-B and Figs 5A-D. The authors should provide a quantitative corresponding definition of "lineage" in the text.

Figure 4. It should be indicated in the figure text that the "lineage" colors in Figs 4A-B correspond to the branch color coding in Figs 3A-B (as above), so that the text of this figure can be read independently. Temporal and spatial distance axis units should be defined in the figure text.

Figure 5. It should be indicated in the figure text that the "lineage" colors in Figs 5A-D correspond to the branch color coding in Figs 3A-D (as above) so that the text of this figure can be read independently.

Supplemental Figure 4. Temporal and spatial axis units should be added to the figure text.

Response to referees for manuscript NCOMMS-19-09762-T, “**Transmission dynamics of multidrug resistant bacteria on intensive care unit surfaces**” by Alaric W. D’Souza, Robert F. Potter, Meghan Wallace, Angela Shupe, Sanket Patel, Xiaoqing Sun, Danish Gul, Jennie H. Kwon, Saadia Andleeb, Carey-Ann D. Burnham, and Gautam Dantas.

We thank the referees and others involved in the editorial process, for their time and effort in considering this manuscript and their thoughtful suggestions to help improve it. Our revised submission includes additional analyses, substantive textual edits, and revised and additional figures, and a revised data table. Having addressed all reviewer concerns, our revised manuscript is greatly strengthened, and we hope a stronger candidate for publication in *Nature Communications*.

We respond to individual comments from referees below:

Responses to Reviewer 1.	
The manuscript “Transmission dynamics of multidrug resistant bacteria on intensive care unit surfaces” describes cultivable contaminants on ICU surfaces in selected hospitals in two countries with substantially different infection control standards. Contaminating microbes on surfaces in US hospitals have been well studied, and little new was found here. Contaminating microbes on surfaces of hospitals in Pakistan were much more abundant and diverse, which was largely predictable, but the specific identification of these microbes and the appearance and disappearance of specific lineages among them is useful for future development of infection control measures that are optimized for this environment. The genetic analysis of the microbes recovered goes farther than most other reports, and is done in a complete and competent manner, providing insights into the ebb and flow of microbes mainly in the Pakistan hospital environment, and a complete inventory of the antibiotic resistances that they harbor.	We thank the reviewer for their kind words about the manuscript. We agree that having specific identifications for microbes is important for infection control. We are also happy the reviewer feels our analysis is “complete and competent” and we hope the reviewer finds our additional analysis and revisions similarly informative, useful, and competent.
Although the broad distribution of antibiotic resistances among various species is known, this study adds new information on their distribution among the organisms isolated. The study observed the co-occurrence of E. faecium with A. baumannii, and explored the ability of the two microbes to form interspecies biofilms. These were found to be robust leading to greater growth and persistence, supporting the prospect that a biological synergy may account for their co-occurrence on contaminated surfaces. While interesting, the studies were somewhat superficial and did not identify anything particularly proprietary about these interactions, or identify such interactions within biofilms recovered from hospital surfaces. Alternatively, it seems plausible	This is an excellent point. It is possible that the bacteria exist on the surfaces in polymicrobial communities, but only A. baumannii and E. faecium survive long term. Additionally, it is possible that our sampling or culturing practices are biased towards yielding A. baumannii and E. faecium. We attempt to forthrightly address this limitation in our revised submission (lines 440-442). Several factors help to mitigate this concern. The first is prior literature we cite showing these bacteria have been found together using 16S sequencing, an approach likely to identify less robust microbes. The second mitigating factor is our results described in this manuscript showing synergistic increases in biofilm formation and biomass between these two species. The

if not likely, that as some of the most environmentally robust microbes, A. baumannii and E. faecium may co-occur among the last cultivable microbes on a surface because of their ability to survive as opposed to any particular biological synergy.	agreement between in silico prediction and in vitro experimentation helps inspire confidence in the result.
In summary, this is an unusually solid study of the microbes that contaminate hospital surfaces mainly in Pakistan. While robust with data of unusual depth, the findings of this report, in this reviewer’s estimation, are largely predictable. As new resistances are emerging and spreading in developing economies where hygiene is difficult and antibiotics are abundant, this data however does provide a nice snapshot of the current situation.	We again thank the reviewer for identifying the study as “unusually solid”. To address the concern of predictability, we have added additional text to highlight novel aspects of the study and to explain how the insights from the study contribute to hospital infection control epidemiology (lines 60-62; 64-65; 70-72; 78-80; 85-93; 383-404; 423-431).
The manuscript would benefit from more sharply defining the rationale for the study design, and from making more obvious to the reader what of the many bioinformatic analyses yielded results that break substantial new ground.	We have added additional text to the introduction (lines 60-62; 64-65; 70-72; 78-80; 85-93) and discussion (lines 383-404; 423-431) to further explain the rationale for the study design and we have included an abstract figure (figure 1) to more clearly depict its structure.
Responses to Reviewer 2	
In this work, D’Souza et al study environmental contamination in ICUs in a US hospital and a Pakistani hospital. The authors performed 180 samplings per hospital over the course of a year, followed by bacterial culture and identification of isolates. 295 Isolates were chosen for phenotypic characterization and genomic sequencing. The authors focus on Acinetobacter, Enterococcus, Klebsiella and Pseudomonas for further genomic analysis. The main findings include a remarkable level contamination of surfaces in the Pakistani hospital, and the persistence of clones of Acinetobacter and Enterococcus on these contaminated surfaces. The authors explore a synergistic biofilm hypothesis to explain why both A. baumannii and E. faecium are found together on dirty surfaces.	We thank the reviewer for summarizing our work.
In this work, the authors performed a number of fairly standard analyses on the genomes of the large set of isolate collected, including phylogenetic comparisons and annotation of resistance genes. This manuscript represents clearly a substantial effort for which the authors are to be applauded. The analysis appears to be technically sound (though see below regarding	We thank the reviewer for prompting us to clarify the novelty of our findings. We have added in additional text to the introduction (lines 60-62; 64-65; 70-72; 78-80; 85-93), results (lines 186-233), and discussion (lines 383-404; 423-431) to highlight specific research gaps that we address and novel findings that we make in this study. Thanks to the reviewer suggestion on genetic

genome assembly statistics). However, I believe the authors need to make a much better case for what is novel in their comprehensive data set. The finding of pervasive poor sanitation in the sampled Pakistani ICU is indeed alarming and actionable, but does not itself constitute an advance in biological knowledge.	variants between strains, we have also added additional novel bioinformatics analyses that more deeply and meaningfully model and evaluate relationships between microbial genome variation across space and time. We illustrate these new results in 2 new main figures [figure 5 and figure 6] and one new supplemental figure [supplemental figure 6] on strain genetic variants that leads to new conclusions on potential contamination routes (lines 186-233; 383-404). We hope these revisions and new analyses make a better case for how our study and conclusions constitute advances in biological knowledge.
Nor does the finding that clonal strains of A. baumannii and E. faecium persist together on the contaminated environmental surfaces over time. These two high impact nosocomial pathogens have been studied in numerous works over the past two decades. Though I find the E. faecium and A. baumannii biofilm experiments very interesting, I think they should be better set in the context of the extensive work that has been done on biofilm formation by these two organisms.	We have added the following additional text to the discussion section to better set our findings in the context of biofilm research. Lines 423-431 Biofilm formation is an important component for the pathogenesis of Enterococcus and Acinetobacter^{1,2}. In both organisms, biofilm formation often requires extracellular attaching proteins including LH92_11085 and OmpA in A. baumannii or the Emp pilus in E. faecium³⁻⁵. Variation has been observed among the ability of A. baumannii clinical isolates to form biofilms, but several strains are capable of growing on urinary catheter surfaces⁶. In E. faecium, adaption to a biofilm is associated with a change in the transcriptional program⁷. 16S rRNA sequencing of high-touch surfaces at large public hospitals in Brazil identified both E. faecium and A. baumannii co-localized to the same surface. Despite this observation and the role of individual genes in biofilm formation for both species there is a dearth of relative knowledge on the specific interactions between these two species that may occur in the built environment. Our results complement this prior work by providing both (1) real-world evidence for significant colocalization of multidrug resistant versions of these bacteria on hospital surfaces which represent high-risk as reservoirs for nosocomial infection, and (2) in vitro support for synergistic biofilm formation between these strains of these bacteria where the mechanisms of biofilm formation are well-defined.
The title refers to “transmission dynamics”. I think a good case is certainly made that dirty surfaces contain mixes of nosocomial clones that are clustered in space and persist over time, but I’m not sure I see the link with what would be	In response to this suggestion we have revised our title to “Spatiotemporal dynamics of multidrug resistant bacteria on intensive care unit surfaces”, which we believe better captures the major themes of our study.

called "transmission dynamics" per se, given that mechanism of movement of isolates between surfaces was not investigated, and patient samples were not included. (I do not believe this detracts from the quality of the study; it is a matter of how the scope is accurately conveyed by the title.)	
The methods state that Illumina assembly statistics were quantified with Quast, but I can't find them in the manuscript. Standard assembly quality parameters must be provided for both the Illumina and Nanopore assemblies. This should include genome assembly lengths, N50, number of contigs, mean + SD coverage statistics, etc. This can be done as a supplementary table.	We have included the requested assembly statistics in supplementary table 1.
Line 566 Clonality analysis. Why were the SNP counts for clonality made on Roary recombination-purged core genomes as opposed to whole genome alignments. The methods used by the authors can underestimate actual differences between isolates where deletions, insertions, duplications, etc. have occurred. Additionally and importantly, were indels (both small and large) included in any of the derivative analyses? Only SNPs are mentioned.	To address this comment, we have done a more in-depth analysis on genetic variation in the isolate genomes. First, we mapped quality filtered reads to the type strain assemblies of A. baumannii and E. faecium. We then called pairwise variants between isolates based on the mapping calls. This approach gives a pairwise distance based on both simple and complex genetic variants. In comparing the variant pairwise distances to the core genome SNP pairwise distances, we found a significant strong correlation for both A. baumannii and E. faecium (supplemental figure 6). We also found that the reviewer was correct, and the new variant method has slightly greater genetic spread, giving us higher resolution in our pairwise comparisons. Using this higher resolution, we grouped the isolates into cliques at every unique cutoff level (see the supplementary zip file for a figure explaining the clique grouping process) within lineage restrictions and then overlaid these cliques with the spatial and temporal data to understand their linkage (Figures 5 and 6). This additional analysis helped us better define the possible contamination routes for these surfaces (lines 186-233; 383-404; 501-528; 585-592).
Figure 1A. Given the substantial differences between PAK-H and USA-H, the authors may consider representing this as two organism abundance histograms rather than one.	We have taken the reviewer's suggestion and split the PAK-H and USA-H bacterial abundance data into two separate panels (figure 2A). We also believe that this gives a stronger impression of the differences between the two countries.
Figure 2. I find the organization of the legend with indented items to be confusing, and many of the indented items (disagreements between MALDI and the ANI-based identifications)	We have followed the reviewer's suggestion and moved figure 2 to the supplement (supplemental figure 1). Axis has also been changed to axes in the figure legend (lines 554-558).

appear to be missing or are not visible in the figure to which the legend refers - for instance Acinetobacter soli (1), Pseudomonas sp. s199 (1), unless I am misinterpreting the figure. I would consider placing this figure in the supplementary data. Minor comment: Axis should be axes in the text.	Acinetobacter soli is included in the figure with the appropriate disagreement annotations. Due to unbiased hierarchical clustering of the Hadamard matrix, it does not group with the other bacteria identified as A. baumannii by MALDI-TOF MS. This holds true for Pseudomonas sp. s199 and several other bacteria, and for clustering accuracy we must leave these isolates as placed.
Figure 3. It would be helpful in the figure text to indicate that the branch color coding in the phylogenetic tree in this figure corresponds to what the authors call “lineages” in Figure 4A-B and Figs 5A-D. The authors should provide a quantitative corresponding definition of “lineage” in the text.	We have added the following text to figure 3 to mitigate this concern. Lines 478-479 Tree branches are colored by hierBAPS lineage and these lineages are colored in subsequent figures.
Figure 4. It should be indicated in the figure text that the “lineage” colors in Figs 4A-B correspond to the branch color coding in Figs 3A-B (as above), so that the text of this figure can be read independently. Temporal and spatial distance axis units should be defined in the figure text.	We have added the following text to address these concerns. Lines 484-485 Lineage from BAP (identified in figure 3 by branch color) is indicated in the legend on the left. Lines 490-492 Temporal distances are calculated as +1 for every 2 week span separating isolate collections. Spatial distances are given as +0 if isolates were collected from the same surface and room, +1 if they were collected from the same room, but different surfaces, and +2 if they were collected from different rooms.
Figure 5. It should be indicated in the figure text that the “lineage” colors in Figs 5A-D correspond to the branch color coding in Figs 3A-D (as above) so that the text of this figure can be read independently.	We have added the following text to the figure legend to mitigate this concern. Lines 533-534 Colored annotations are added next to the resistance genes for resistance gene class and above the charts for hierBAPS lineage (identified in figure 3 by branch color), week, surface, and room.
Supplemental Figure 4. Temporal and spatial axis units should be added to the figure text.	We have added the following text to the figure legend to describe the spatial and temporal axis units. Lines 577-579 Temporal distances are calculated as +1 for every 2 week span separating isolate collections. Spatial distances are given as +0 if isolates were collected from the same surface and room, +1 if they were collected from the same room, but different surfaces, and +2 if they were collected from different rooms.

References

1. Mohamed JA, Huang DB. Biofilm formation by enterococci. *J Med Microbiol.* 2007;56(Pt 12):1581-8. Epub 2007/11/24. doi: 10.1099/jmm.0.47331-0. PubMed PMID: 18033823.
2. Wong D, Nielsen TB, Bonomo RA, Pantapalangkoor P, Luna B, Spellberg B. Clinical and Pathophysiological Overview of Acinetobacter Infections: a Century of Challenges. *Clin Microbiol Rev.* 2017;30(1):409-47. Epub 2016/12/16. doi: 10.1128/CMR.00058-16. PubMed PMID: 27974412; PMCID: PMC5217799.
3. Alvarez-Fraga L, Perez A, Rumbo-Feal S, Merino M, Vallejo JA, Ohneck EJ, Edelmann RE, Beceiro A, Vazquez-Ucha JC, Valle J, Actis LA, Bou G, Poza M. Analysis of the role of the LH92_11085 gene of a biofilm hyper-producing *Acinetobacter baumannii* strain on biofilm formation and attachment to eukaryotic cells. *Virulence.* 2016;7(4):443-55. Epub 2016/02/09. doi: 10.1080/21505594.2016.1145335. PubMed PMID: 26854744; PMCID: PMC4871663.
4. Schweppe DK, Harding C, Chavez JD, Wu X, Ramage E, Singh PK, Manoil C, Bruce JE. Host-Microbe Protein Interactions during Bacterial Infection. *Chem Biol.* 2015;22(11):1521-30. Epub 2015/11/10. doi: 10.1016/j.chembiol.2015.09.015. PubMed PMID: 26548613; PMCID: PMC4756654.
5. Montealegre MC, Singh KV, Somarajan SR, Yadav P, Chang C, Spencer R, Sillanpaa J, Ton-That H, Murray BE. Role of the Emp Pilus Subunits of *Enterococcus faecium* in Biofilm Formation, Adherence to Host Extracellular Matrix Components, and Experimental Infection. *Infect Immun.* 2016;84(5):1491-500. Epub 2016/03/02. doi: 10.1128/IAI.01396-15. PubMed PMID: 26930703; PMCID: PMC4862714.
6. Pour NK, Dusane DH, Dhakephalkar PK, Zamin FR, Zinjarde SS, Chopade BA. Biofilm formation by *Acinetobacter baumannii* strains isolated from urinary tract infection and urinary catheters. *FEMS Immunol Med Microbiol.* 2011;62(3):328-38. Epub 2011/05/17. doi: 10.1111/j.1574-695X.2011.00818.x. PubMed PMID: 21569125.
7. Lim SY, Teh CSJ, Thong KL. Biofilm-Related Diseases and Omics: Global Transcriptional Profiling of *Enterococcus faecium* Reveals Different Gene Expression Patterns in the Biofilm and Planktonic Cells. *OMICS.* 2017;21(10):592-602. Epub 2017/10/20. doi: 10.1089/omi.2017.0119. PubMed PMID: 29049010.

REVIEWERS' COMMENTS:

Reviewer #2 (Remarks to the Author):

I think the authors have adequately addressed all of my comments in this version of the manuscript. The inclusion of the additional graph-based analysis of the culture data is interesting and useful, and adds an additional dimension of novelty to this work. In summary, I believe this represents an unusually comprehensive study of nosocomial contamination of hospital surfaces and will be a significant contribution to the literature. I have no further comments or concerns.

John Dekker